# Through the Lens: Benchmarking Deepfake Detectors Against Moiré-Induced Distortions

**Razaib Tariq**[*]      **Minji Heo**[*]      **Simon S. Woo**[1][†]          **Shahroz Tariq**[†]
Sungkyunkwan University, South Korea                    CSIRO's Data61, Australia
[1] Secure Machines Lab Inc, South Korea               `shahroz.tariq@data61.csiro.au`
`{razaibtariq,minji.h0224,swoo}@g.skku.edu`

## Abstract

Deepfake detection remains a pressing challenge, particularly in real-world settings where smartphone-captured media from digital screens often introduces Moiré artifacts that can distort detection outcomes. This study systematically evaluates state-of-the-art (SOTA) deepfake detectors on Moiré-affected videos, an issue that has received little attention. We collected a dataset of 12,832 videos, spanning 35.64 hours, from the Celeb-DF, DFD, DFDC, UADFV, and FF++ datasets, capturing footage under diverse real-world conditions, including varying screens, smartphones, lighting setups, and camera angles. To further examine the influence of Moiré patterns on deepfake detection, we conducted additional experiments using our DeepMoiréFake, referred to as (DMF) dataset and two synthetic Moiré generation techniques. Across 15 top-performing detectors, our results show that Moiré artifacts degrade performance by as much as 25.4%, while synthetically generated Moiré patterns lead to a 21.4% drop in accuracy. Surprisingly, demoiréing methods, intended as a mitigation approach, instead worsened the problem, reducing accuracy by up to 17.2%. These findings underscore the urgent need for detection models that can robustly handle Moiré distortions alongside other real-world challenges, such as compression, sharpening, and blurring. By introducing the DMF dataset, we aim to drive future research toward closing the gap between controlled experiments and practical deepfake detection.

## 1   Introduction

The rise of deepfake technology has transformed how digital media can be manipulated, presenting a growing threat across the internet and social networking platforms. Deepfakes, which are artificially generated or altered videos that convincingly imitate real individuals, pose significant risks to privacy, security, and the spread of misinformation. The increasing ease with which deepfakes can be created exacerbates this issue [1], as their realism often deceives the general public and sophisticated detection algorithms. Advances in deepfake generation techniques, such as those using Generative Adversarial Networks (GANs) [2] and other deep learning models [3, 4], including diffusion models [5], have made detection an extremely challenging task [6–9] in real-world scenarios on the Internet. While efforts have been made to develop robust detection systems [10–21], such algorithms are predominantly evaluated in controlled environments using benchmark datasets. However, real-world scenarios introduce various challenges, including environmental factors and media-sharing distortions, which can significantly impact detection accuracy [22]. One of the most prominent challenges arises when deepfake content is viewed on screens and recorded using smartphone cameras. Although naive screen capture is available, Digital Rights Management (DRM) on many platforms often disables it.

---

[*]Equal contribution.
[†]Corresponding author.

39th Conference on Neural Information Processing Systems (NeurIPS 2025) Track on Datasets and Benchmarks.

Accordingly, we focus on the prevalent smartphone screen-recapture scenario that users adopt for casual sharing or unauthorized reproduction. In practice, the same deepfake can exhibit drastically different visual characteristics when viewed directly on a screen versus when captured by a camera, adding an extra layer of complexity for detection systems (See Figure 1). This common real-world scenario introduces visual artifacts known as Moiré patterns, which occur due to the interference between the pixel grid of the display and the camera sensor [23]. These Moiré patterns, often undetected by the human eye, severely disrupt deepfake detection algorithms, highlighting a critical gap between controlled environment performance and practical, real-world conditions.

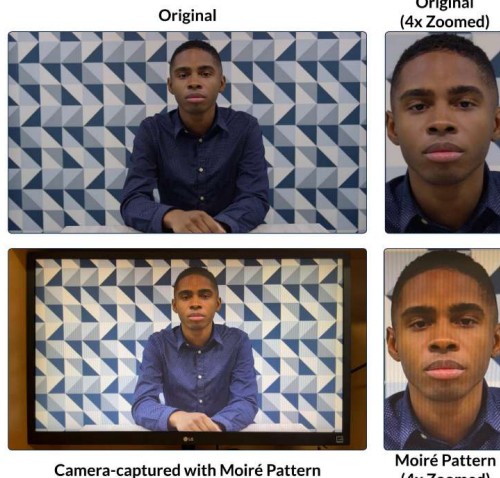

**Original**    **Original (4x Zoomed)**

**Camera-captured with Moiré Pattern**    **Moiré Pattern (4x Zoomed)**

Figure 1: **Original vs. Moiré pattern**

In this paper, we investigate the impact of Moiré patterns and compression on deepfake detection systems across three scenarios: (i) Authentic Moiré patterns, (ii) Synthetic Moiré patterns, and (iii) Compression Attacks. Authentic Moiré patterns are introduced when users record content displayed on the screen with smartphones, degrading detection accuracy by distorting key visual features [24]. For instance, a deepfake video of President Putin was shown nationwide on television declaring martial law, which was captured with a smartphone, clearly showing signs of a Moiré pattern [25, 26] shared on X (formerly Twitter). Another example of a smartphone-captured deepfake on social media is creating false narratives by a broadcaster announcing President Macron rescheduling a visit due to an assassination attempt [27]. Synthetic Moiré patterns, on the other hand, are deliberately generated either through pixel-level manipulation or by capturing screen-displayed content with controlled distortions, obscuring the artifacts that deepfake detectors rely on. Finally, Compression Attacks simulate real-world video uploads to social networking sites (SNS), where compression artifacts combine with Moiré patterns to impair detection systems further.

We address these challenges by introducing the DMF dataset, the first deepfake dataset to incorporate Moiré patterns into public deepfake datasets. It includes diverse videos captured from four screens under two lighting conditions with two smartphones, providing a realistic

Table 1: **DeepMoiréFake Details:** We selected a subset of videos from five famous deepfake datasets and manually captured them under various conditions, which resulted in a total playback time of 35.64 hours containing Moiré patterns across 12,832 videos (802×4 (screens)×2 (phones)×2 (lightning conditions).

| NAME | REAL VIDEOS (People) | FAKE VIDEOS (People) | VIDEOS FROM DATASET | DURATION PER VIDEO (secs) | TOTAL VIDEOS CAPTURED | CAPTURED DURATION (hours) |
|---|---|---|---|---|---|---|
| FF++ [28] | 200 | 200 | 400 | 10 | 6400 | 17.78 |
| DFD [29] | 28 | 28 | 56 | 10 | 896 | 2.49 |
| DFDC [30] | 66 | 66 | 132 | 10 | 2112 | 5.87 |
| CelebDF [31] | 58 | 58 | 116 | 10 | 1856 | 5.16 |
| UADFV [32] | 49 | 49 | 98 | 10 | 1568 | 4.36 |
| **Total** | **401** | **401** | **802** | - | **12832** | **35.64** |

benchmark for evaluating the resilience of state-of-the-art deepfake detectors. Table 1 and Table 2 detail the dataset's variations, and video-capturing specifications, reflecting practical challenges in deepfake detection. Unlike our previous studies [33] and [34], this work adds details not covered before. Specifically, we present a broader analysis of deepfake detectors, examine distortions beyond Moiré patterns, and evaluate compression effects and mitigation strategies in depth. These differences are discussed in Section 2, and the various angles used in our analysis are outlined in Appendix P. To assess the effectiveness of our dataset, we performed an extensive evaluation using 15 different deepfake generation methods. Additionally, we examine the impact of defense methods such as demoiréing techniques on the performance of these detection algorithms as a potential mitigation strategy. We summarize our main contributions as follows:

1. **Moiré Pattern Attacks, Scenarios, and Datasets:** We propose Authentic Moiré patterns and Synthetic Moiré patterns. Constructively, we developed the first Moiré Pattern-impacted deepfake datasets to evaluate both real-world cases. They are captured with four different computer screens using two different smartphone cameras under two different lighting conditions on videos from

FaceForensics++ (FF++) [28], CelebDF [31], the DeepFake Detection (DFD) [29], the DeepFake Detection Challenge (DFDC) [30] and UADFV [32] dataset. DMF is released publicly under DOI-based restricted terms and conditions to support further research on Moiré-induced challenges in deepfake detection[3], and our evaluation codes are publicly available here[4].

2. **Extensive Moiré Pattern Evaluation and Benchmarking:** We conducted an extensive empirical study using our DMF dataset and 15 detectors to determine how Moiré patterns from camera-captured deepfake videos on digital screens affect detector performance. This helps in understanding real-world application challenges and vulnerabilities with current detection methods.

3. **Mitigation and Defense Approach with Demoiréing.** To remove the Moiré pattern from DMF videos, we propose the state-of-the-art defense methods and apply them using five image and two video demoiréing methods, evaluate these demoiréd videos using the identical 15 deepfake detectors, and present the effectiveness and implications of defense methods.

Our real-world evaluation revealed that the presence of Moiré patterns caused an average performance decline of 10.7% in deepfake detectors, with reductions reaching up to 25.4% in extreme cases. Additionally, implementing demoiréing as a defense further decreased detection accuracy, with an average decline of 6.1% and up to 17.2% in severe cases. These findings highlight the need for further research to understand better the interaction between demoiréing techniques and deepfake detection algorithms.

Table 2: Specifications and variations in the video-capturing setup.

| NAME (VARIATIONS) | DETAILS |
|---|---|
| CAMERA ANGLES (4) | Center, 45° left, 45° right and Handheld |
| LIGHTNING CONDITIONS (2) | On and Off |
| SCREENS (4) (60 (Hz)) | LG (LED), BenQ (LED), Samsung (QHD-IPS), and Lenovo (UHD-IPS) |
| PHONES (2) | iPhone 13 Samsung S22 Plus |
| SCREEN RESOLUTION (2) | 1980x1080, 3840x2160 |
| CAPTURE RESOLUTION (1) | 1980x1080 |
| FRAME RATE (1) | 30 fps |
| VIDEO CAPTURE APPS (2) | OBS Studio, DroidCam (iOS) IP Webcam (Android) |

## 2 Related Work

**DEEPFAKE GENERATION.** Deepfake video generation leverages advanced deep learning techniques such as variational autoencoders (VAEs) [35], generative adversarial networks (GANs) [2], and diffusion models [36] to produce highly realistic manipulated videos. Common deepfake manipulations include face swapping, face reenactment, face attribute editing, and face synthesis [37, 38]. *Face swapping* replaces a target face with a source face while preserving attributes such as skin color, expressions, and the surrounding environment [3]. *Face reenactment* transfers expressions and movements from a source face to a target, retaining the target's appearance and identity. This technique uses facial motion capture and deep learning to modify the target's movements based on a driving image, video, or pose [39–41]. *Face attribute* editing alters specific facial features, such as age, expressions, or skin tone, using generative models among GANs and VAEs. It can focus on single attributes or edit multiple attributes simultaneously [42, 43]. Finally, *face synthesis* employs GANs to create hyper-realistic human faces that do not exist. While it has applications in gaming and fashion, it also poses risks, such as enabling fake identities on social networks to spread misinformation [44, 45].

**DEEPFAKE DETECTION.** As deepfake generation technology advances, effective detection methods become increasingly critical to prevent misuse. Deepfake detection relies on deep learning models that identify subtle artifacts often imperceptible to the human eye. Techniques include convolutional neural

Table 3: A comparison of publicly available Deepfake datasets.

| DATASET | REAL VIDEOS | FAKE VIDEOS | TOTAL VIDEOS | ENCODING ARTIFACTS | ACQUISITION ARTIFACTS |
|---|---|---|---|---|---|
| UADFV [32] | 49 | 49 | 98 | ✗ | ✗ |
| DeepfakeTIMIT [46] | 640 | 320 | 960 | ✓ (Compress.) | ✗ |
| FF++ [28] | 1,000 | 4,000 | 5,000 | ✓ (Compress.) | ✗ |
| CelebDF [31] | 590 | 5,639 | 6,229 | ✗ | ✗ |
| DFD [29] | 363 | 3,000 | 3,363 | ✗ | ✗ |
| DeeperForensics [47] | 50,000 | 10,000 | 60,000 | ✗ | ✗ |
| DFDC [48] | 23,654 | 104,500 | 128,154 | ✓ (Compress.) | ✓ (Lighting) |
| KoDF [49] | 62,166 | 175,776 | 237,942 | ✗ | ✗ |
| FakeAVCeleb [50] | 500 | 19,500 | 20,000 | ✗ | ✗ |
| **Ours** | **401** | **401** | **802** | ✗ | ✓ **(Moiré)** |

[3]https://doi.org/10.7910/DVN/XYOSYW
[4]https://github.com/Razaib-Tariq/DeepMoireFake

networks [51–56], temporal analysis [57], frequency domain analysis [19], and attention mechanisms using transformers [58–60]. Detection methods are typically developed and evaluated using datasets such as FaceForensics++ (FF++) [28], CelebDF [31], UADFV [32], and FakeAVCeleb [50]. A comparative analysis of existing datasets is presented in Table 3.

Our work evaluates deepfake detection systems under three distinct real-world scenarios. Captured Moiré Pattern Attack (CMPA) simulates authentic Moiré patterns generated when users record deepfake videos from screens, causing distortions that obscure critical visual features and degrade detection accuracy. Synthetic Moiré Pattern Attacks (SMPA) investigate the effects of artificial Moiré patterns using methods such as SMPA-MA [61] and SMPA-SPS [62]. Finally, Compression Attack (CA) explores how video compression artifacts from SNS uploads interact with Moiré patterns, further degrading deepfake detection performance.

**MOIRÉ PATTERNS IN DEEPFAKE DETECTION.** While deepfake detection has seen significant advancements, the impact of Moiré patterns on detection performance remains an underexplored challenge. Prior studies, including our previous works [33] and [34], have investigated deepfake detection under various conditions; however, these efforts were limited in scope. The former employed a restricted set of detection methods, offering only preliminary insights on a constrained dataset, whereas the latter expanded the evaluation to more detectors but lacked real-world scenarios where Moiré patterns could be actively exploited. Furthermore, these studies did not systematically assess critical factors such as image distortion, compression effects, and mitigation strategies, essential for improving robustness in practical applications.

In contrast, our study extensively explores these missing aspects and introduces the novel DMF dataset to fill these gaps. The dataset comprises videos captured from four screens under two lighting conditions using two smartphones, enabling robust and practical evaluations. By capturing real-world interference patterns under controlled variations, the DMF dataset enables robust and practical evaluations, providing a more comprehensive benchmark for deepfake detection. Moreover, we analyze both authentic and synthetic Moiré patterns, extending prior research [33] and [34]. Beyond this, we systematically investigate mitigation strategies, including demoiréing, denoising, and deblurring, revealing significant trade-offs where removing Moiré patterns may inadvertently reduce detection accuracy. Our empirical analysis spans 15 deepfake detectors and rigorously evaluates the interplay between Moiré patterns and compression artifacts, providing a robust real-world assessment.

## 3 Dataset Collection and Generation

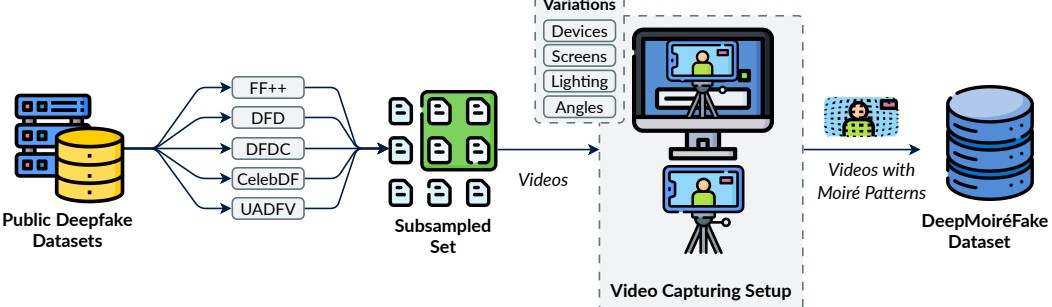

Figure 2: Our manual Moiré pattern collection pipeline.

**SELECTION AND COLLECTION.** We selected five public deepfake datasets, UADFV [32], FaceForensics++ [28], DFD [29], DFDC [30], and CelebDF [31] as a representative set, covering a variety of settings detailed in Table 3. The overall process of generating our DeepMoiréFake dataset from these source datasets is illustrated in Figure 2, which shows the pipeline of subsampling videos and capturing them under various screen and device conditions to induce Moiré patterns. UADFV [32] dataset was created using the FakeApp tool and contains 49 real YouTube videos, each paired with a corresponding deepfake video. Videos are approximately 11 seconds long, with a resolution of 294×500 pixels. FaceForensics++ (FF++) [28] dataset includes 1,000 real YouTube videos and 1,000 deepfake videos generated using four techniques: Deepfake [4], Face2Face [40], Faceswap [3], and NeuralTexture [39]. It provides 4,000 manipulated videos in three quality levels uncompressed (raw),

lightly compressed (C23), and highly compressed (C40), enabling evaluations across varying compression levels. DFD [29] dataset by Google/Jigsaw consists of 3,068 deepfake videos generated from 363 original videos of 28 consented individuals representing diverse genders, ages, and ethnicities. DFDC [30] dataset is the largest public faceswap dataset, featuring over 100,000 videos from 3,426 paid actors. Most videos are in 1080p resolution and include a mix of deepfake, GAN-based, and non-learned techniques, with an average of 14.4 videos per individual. CelebDF [31] dataset contains 590 real and 5,639 deepfake videos, sourced from over two million frames of YouTube interviews with 59 celebrities of diverse genders, ages, and ethnicities.

**DATASET SUBSET SELECTION.** We initially selected a subset of videos from five public deepfake datasets for dataset generation, ensuring a balanced representation across gender and ethnicity. Firstly, from the FaceForensics++ (FF++) dataset, we randomly chose 50 real and 50 fake videos from each of the four sub-datasets, totaling 400 videos. For the DFD, DFDC[5], CelebDF, and UADFV datasets, we selected one real and one fake video for each unique individual, resulting in 56, 132, 116, and 98 videos, respectively (see Table 1), resulting in a carefully selected 802 videos from these datasets. Our main objective was to ensure a diverse representation from these deepfake datasets while maintaining a manageable number of manual videos for manual handling when creating Moiré pattern videos for our DMF dataset.

**GENERATION OF DEEPMOIRÉFAKE (DMF) DATASET.** The DMF dataset addresses the limitations of existing deepfake datasets by replicating real-world conditions, where videos are captured on monitor screens using mobile devices. This approach introduces distortions, such as Moiré patterns and screen-specific characteristics, which are captured through mobile device cameras, thereby facilitating the evaluation of deepfake detection methods in practical scenarios. The dataset includes real and deepfake content recordings displayed on four monitors using two smartphone cameras (iPhone 13 and Samsung S22 Plus). Each smartphone was positioned on a stand 35 cm from the screen at various angles. Videos were captured under varied lighting conditions, with specifications detailed in Table 2. This carefully designed setup ensures a diverse range of screen types, lighting conditions, and device configurations, making the dataset valuable for advancing deepfake detection in real-world applications. During this stage, we ensured label accuracy via an automated process followed by manual verification and enforced consistency across screen camera setups. We also categorized each sample by device type, display screen, viewing angle, and lighting environment. These attributes are recorded in the dataset metadata to support reproducibility and downstream analysis.

**DEEPFAKE DETECTION.** To evaluate deepfake detection performance across different datasets, we utilized 15 deepfake detectors. Specifically, we employed 10 image-based detectors, including SelfBlended [63], Rossler [28], ForgeryNet [64], Capsule-Forensics (Capsule) [65], MAT [66], CADDM [67], CCViT [58], and ADD [19] to assess detection results on the original dataset, Moiré pattern dataset, and demoiréd dataset. For the Rossler [28], we used pre-trained weights from three variations of the FaceForensics++ dataset: raw, C23, and C40, referred to as Rossler, Rossler C23, and Rossler C40, respectively. In our deepfake video detection experiments, we employed 5 detectors: AltFreezing [68], FTCN [69], LRNet [70], and LipForensics [71]. For LRNet, we used the BlazeFace (LRNet BF) and RetinaFace (LRNet RF) variants.

## 4 Experimental Scenarios and Settings

**Authentic Moiré Patterns or Captured Moiré Pattern Attack (CMPA).** To simulate and evaluate a user-generated distortion, we propose a ***Captured Moiré Pattern Attack (CMPA)***. Moiré patterns, caused by interference between the pixel grids of the camera and monitor display [72], are more intense with more significant resolution mismatches and are particularly pronounced on older technologies such as LCD, LED, and IPS displays compared to OLEDs. This effect diminishes as the distance between the camera and monitor increases [73]. These distortions degrade deepfake detection accuracy by introducing artifacts that interfere with identifying critical visual features, especially over multiple frames. Deepfakes were generated using a range of monitors with varying resolutions, such as 1080p LED, 1440p QHD IPS, and 4K UHD IPS displays, to simulate real-world conditions. Future datasets should include OLED displays and 8K monitors to improve the robustness of detection algorithms against these evolving challenges [74].

---

[5]For DFDC, we selected videos from the preview version, containing 5000 videos of 66 unique individuals.

**SYNTHETIC MOIRÉ PATTERN ATTACKS (SMPA).** To evaluate the impact of interference patterns on deep learning models, we propose *Synthetic Moiré Pattern Attacks (SMPA)*, which replicate noise artifacts commonly observed in screen recordings (see Figure 3). These patterns degrade model performance by introducing complex distortions that are difficult to detect and eliminate. The SMPA approach we used incorporates two methods: (1) **SMPA-MA**, which simulates real-world capture conditions by applying scaling, resampling, and random rotations to input images, as proposed by [61], and (2) **SMPA-SPS**, which modulates parameters such as skew, contrast, and deviation while introducing non-linear distortions such as sine waves to replicate complex Moiré patterns [62]. By mimicking real-world Moiré artifacts, the SMPA demonstrates an efficient and effective adversarial approach, emphasizing the importance of designing robust detection algorithms to counter such attacks.

**COMPRESSION ATTACK WITH CMPA AND SMPA.** Uploading videos to Social Networking Sites (SNS) often introduces compression and quality degradation, adding new artifacts to the content. To replicate real-world scenarios, we propose the *Compression Attack (CA)* on Moiré Patterns, which combines Moiré distortions with compression artifacts to simulate the impact of social media uploads. This approach leverages the widely used H.264 compression algorithm, adopted by platforms such as TikTok and YouTube [75], and standard techniques from FaceForensics++ [28]. By mirroring these real-world compression methods, we provide a realistic evaluation of how compression affects deepfake detection performance. We generated two compressed versions of our dataset, C23 and C40, to simulate the quality degradation caused by SNS uploads [76, 77]. Compression reduces high-frequency information, introducing noise that interacts with existing Moiré patterns to create more complex distortions. These combined artifacts significantly degrade deepfake detection systems' performance, with increasing compression noise leading to further reduc-

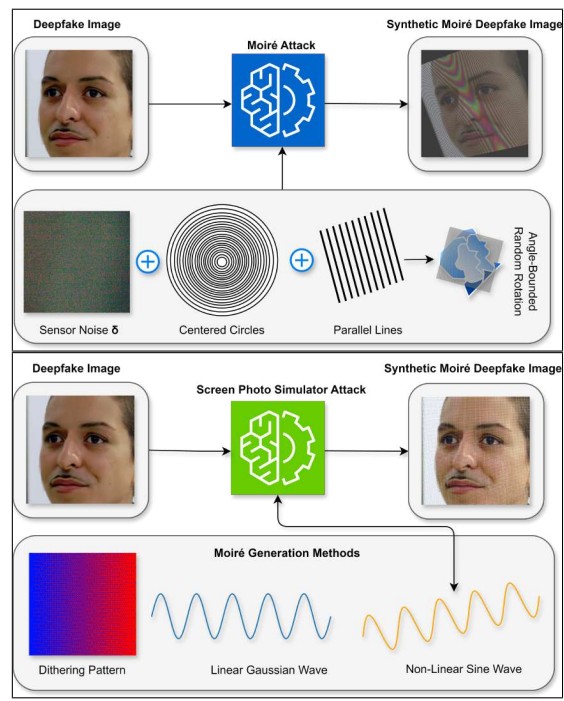

Figure 3: Synthetic Moiré Generation

tions in detection accuracy. This attack reveals a critical vulnerability in current detection systems, which often overlook the combined effects of Moiré patterns and compression artifacts. By exploiting compression, attackers can obscure signs of manipulation, enabling altered videos to evade detection on SNS platforms.

**PREPROCESSING AND PRETRAINED WEIGHTS.** Our dataset comprised videos from various sources, including CelebDF [31], DFD [29], DFDC [30], and FaceForensics++ [28]. Each dataset was preprocessed according to the specific requirements of the respective deepfake detectors. For the demoiré experiments, an additional preprocessing step was applied to remove the Moiré pattern using state-of-the-art demoiréing methods [78–81]. We used the pretrained weights for each detector during the evaluation. We selected the top five performers on CMPA from the image detectors Rossler C23, MAT, CADDM, SelfBlended, and CCViT for SMPA, CA, demoiréing, fine-tuning, and retraining experiments. However, SelfBlended and CCViT did not exhibit any performance improvement during training, remaining at a static accuracy of 50%. As a result, we excluded them from further analysis and focused on Rossler C23, MAT, and CADDM.

**MOIRÉ REMOVAL METHODS.** We employed state-of-the-art demoiréing methods such as DMCNN [78], MBCNN [79], and ESDNet [80] (under two settings) alongside DDA [81], which is tailored for mobile devices, to remove the Moiré pattern from DMF videos: Firstly, (i) **DMCNN** utilizes groups of convolutional layers for downsampling and deconvolutional layers to restore resolution. The final output image is produced by summing feature maps from all branches; Secondly, (ii) **MBCNN** features a learnable bandpass filter (LBF) for effective Moiré texture removal and employs a two-step tone

mapping strategy for color restoration. This includes global tone mapping to correct color shifts and local fine-tuning for per-pixel accuracy. Thirdly, (iii) **ESDNet** integrates a Semantic-Aligned Scale-Aware Module (SAM) to handle scale variations of Moiré patterns. It enhances model effectiveness by extracting and dynamically fusing multi-scale features within the same semantic level, maintaining a lightweight network structure. and Lastly, (iv) **DDA** is optimized for real-time deployment on mobile devices. This method employs a parameter-shared supernet paradigm, ensuring resource efficiency without adding an extra parameter burden. It was selected because our data collection was performed using mobile devices. The demoiréing experiments were conducted on image and video-based techniques, which are provided in the Appendix Table 11 and Table 12.

**METRICS.** We evaluated the performance of deepfake detectors in our experiments using accuracy, AUC score, precision, recall, and F1-score. The main text reports the results based on the AUC score. For the CA, fine-tuning, and retraining settings, we report the best Accuracy. The results for the other metrics are available in the Appendix H.

## 5 Results

**CMPA – PERFORMANCE UNDER VARIOUS PLAYBACK SCREEN SETTINGS.**

In Table 4, the most significant visual Moiré artifacts were observed when videos were captured from the LG and BenQ screens, both of which use backlit LED technology with low pixel density and traditional RGB stripe subpixel layouts. These structural characteristics tend to amplify aliasing effects, particularly when captured through camera sensors, resulting in severe Moiré distortions. Correspond-

Table 4: Performance on different playback screens.

| DETECTORS | | ORIGINAL | Videos captured from screens | | | |
|---|---|---|---|---|---|---|
| (Type and Name) | | PERFORMANCE | LG | BenQ | Lenovo | Samsung |
| VIDEO | LRNet BF | 61.7 | 54.9 | 55.3 | 55.9 | 53.2 |
| | LRNet RF | 62.2 | 58.8 | 60.5 | 58.7 | 58.8 |
| | FTCN | 90.2 | 65.9 | 65.3 | 70.6 | 68.9 |
| | LipForensics | 90.6 | 80.3 | 80.8 | **84.4** | 79.8 |
| | AltFreezing | **92.5** | **80.4** | **81.3** | 83.7 | **82.9** |
| IMAGE | Rossler | 67.7 | 56.2 | 54.5 | 59.4 | 56.9 |
| | ADD | 69.7 | 65.4 | 64.3 | 66.3 | 63.4 |
| | Capsule | 71.3 | 71.2 | 69.6 | 69.0 | 66.6 |
| | ForgeryNet | 76.9 | 61.5 | 61.8 | 66.5 | 63.6 |
| | Rossler C40 | 77.0 | 67.7 | 66.9 | 67.3 | 67.8 |
| | Rossler C23 | 86.5 | 68.6 | 67.4 | 74.5 | 70.9 |
| | MAT | 87.0 | 72.4 | 74.9 | 80.1 | 76.6 |
| | CADDM | 87.1 | 71.3 | 71.8 | 80.9 | 79.5 |
| | SelfBlended | 88.8 | 73.7 | 75.5 | 80.9 | 76.4 |
| | CCViT | **95.0** | **81.9** | **83.7** | **86.4** | **86.0** |
| **Avg. Performance loss (Moiré vs. Original)** | | | -11.6 | -11.4 | -8.0 | -10.2 |

ingly, the most substantial performance degradation in detection was also recorded for these two screens. This indicates that certain display technologies might amplify Moiré artifacts more than others. The variations in pixel arrangements, refresh rates, and anti-aliasing techniques across different screens likely contribute to the severity of these distortions. CCViT [58] demonstrated the best detection performance across all screen environments, with an average AUC of 84.5%. Meanwhile, Capsule [65] and LRNet showed robustness in this with the different capturing devices scenarios, and performance dropping by only 2-3 percentage points, for instance, on average, Rossler C23 [28] performance dropped to 16.1%, whereas Capsule experienced only a 2.2% drop. The performance from Capsule and LRNet is significantly low (around the mid-60s), making them impractical in the real world. Overall, we observed a similar trend in performance results across different screen configurations. In addition, we include performance results on videos captured at ±45° viewing angles in the Appendix Table 15, further examining how angled perspectives affect detection robustness under Moiré interference.

**CMPA – PERFORMANCE WITH DIFFERENT CAPTURING DEVICES.** In Figure 4, we illustrate the performance of detectors with original and Moiré pattern captured videos using iPhone and Samsung devices, showing a significant performance drop, highlighting the impact of Moiré artifacts on deepfake detection. The detection performance on videos captured using the Samsung S22 Plus was slightly worse on average than that captured with the iPhone. CCViT [58] achieved the best performance across all scenarios, with 95% on the original, 85% on iPhone-captured, and 83% on Samsung-captured images. The worst performance was observed with the LRNet models and Rossler model [28], where Rossler scored 68% on the original, 58% on iPhone-captured, and 55% on Samsung-captured images, suggesting that the severity of Moiré interference may vary across different smartphone camera sensors and image processing pipelines. Overall, all detectors have a significant drop in performance, regardless of the capturing device used. This consistent degradation

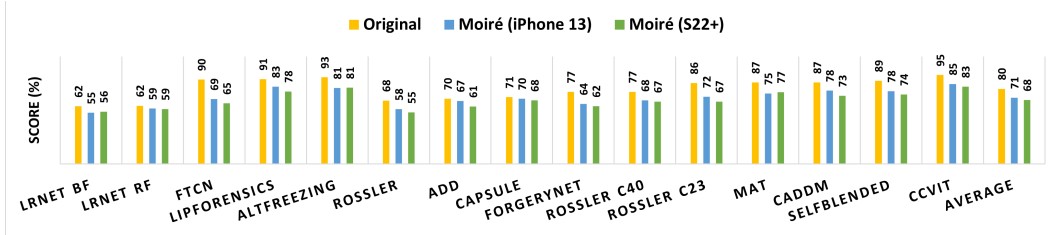

Figure 4: **Different Capturing Devices:** AUC performance of detectors dropped by 9.5 and 12.0 percentage points on average for videos with Moiré patterns captured by iPhone 13 and Samsung S22 Plus, with a maximum drop of 25.4 percentage points in the worst case.

raises concerns about the generalizability of deepfake detectors in real-world scenarios where Moiré artifacts are commonly introduced during video playback or screen recording.

**CMPA – Performance Under Different Lighting Conditions.** The Table 5 shows the performance of the (top-3) detectors when videos are captured by the camera on different screens that are exposed to different lighting conditions. In this scenario, we observed a very minimal performance change (around 1%), which shows that the impact of the Moiré pattern remained the same irrespective of the lighting conditions (see Appendix Figure 14 for results of all detectors).

Table 5: Lighting Conditions

| Detectors | On | Off |
|---|---|---|
| LipForensics | 81.3 | 78.7 |
| AltFreezing | 82.1 | 80.6 |
| CCViT | 84.5 | 83.5 |

**SMPA – Synthetic Moiré Pattern Attacks Results.** We examined two types of Synthetic Moiré Attacks. One is SMPA-MA, and the other is SMPA-SPS. Each Synthetic Moiré Attack framework is shown in (Figure 3). By using a subset of one variation of the camera-captured videos. MAT shows the most severe performance degradation by synthetic Moiré attack among three detector models, with a performance drop of 21.4 percentage points (see Table 6). Unlike MAT, which shows performance degradation after the Synthetic Moiré Attack, Rossler and CADDM show improved performance after SMPA-MA.

Table 7: CA baseline results under C23 and C40, evaluated with each detector's pretrained weights.

| Detectors | C23 | | | | C40 | | | |
|---|---|---|---|---|---|---|---|---|
| | OG | CMPA | SMPA-MA | SMPA-SPS | OG | CMPA | SMPA-MA | SMPA-SPS |
| Rosseler C23 | **98.4** | **96.5** | **87.7** | 80.5 | **87.5** | **99.3** | 83.2 | **98.7** |
| MAT | 86.7 | 66.1 | 55.4 | 56.5 | 75.3 | 66.5 | 52.2 | 60.8 |
| CADDM | 97.7 | 96.4 | 86.7 | **90.3** | 80.1 | 99.0 | **84.8** | 96.8 |

Table 8: Performance of fine-tuned and retrained models on C23 and C40 compression attacks.

| | Detectors | Fine-tune | | | | Retrain | | | |
|---|---|---|---|---|---|---|---|---|---|
| | | OG | CMPA | SMPA-MA | SMPA-SPS | OG | CMPA | SMPA-MA | SMPA-SPS |
| C23 | Rossler C23 | 98.0 | **96.5** | 88.6 | 91.0 | 97.8 | 96.1 | 87.8 | 91.5 |
| | MAT | 99.2 | 91.8 | 94.8 | **98.5** | 99.3 | 92.1 | **95.8** | **97.5** |
| | CADDM | **99.8** | 96.3 | **95.0** | 92.0 | **99.4** | **96.2** | 90.0 | 91.8 |
| C40 | Rossler C23 | 82.5 | **99.6** | 85.7 | 97.0 | 86.7 | 99.5 | 85.7 | 95.7 |
| | MAT | **98.0** | 90.6 | **94.4** | 97.9 | **99.1** | 84.2 | **94.4** | 98.1 |
| | CADDM | 90.9 | 99.3 | 90.8 | **99.2** | 96.0 | **99.7** | 90.71 | **99.3** |

**Compression Attacks (CA).** In Table 7 we observe that for the CA baseline, methods such as Rossler C23, MAT, and CADDM show distinct accuracy ranges, with Rossler C23 and CADDM achieving around 80.1–99.0% and MAT lagging behind at 55.2–86.7%. Following fine-tuning and retraining, however, the overall trend is upward, as detailed in Table 8. Most notably, the MAT model's accuracy surged to 99.2% in the best case, effectively closing the gap and becoming competitive with the other methods. This

Table 6: Comparison of deepfake detector performance in the presence and absence of Moiré Attacks.

| Detectors | Without Attack | Moiré Attack | | |
|---|---|---|---|---|
| | | CMPA | SMPA-MA | SMPA-SPS |
| Rossler C23 | **78.1** | **81.9** | 83.1 | 75.4 |
| MAT | 76.8 | 68.8 | 55.4 | 61.8 |
| CADDM | 73.0 | 73.1 | **86.8** | **80.7** |

indicates that fine-tuning or retraining models on specific datasets or with targeted adjustments can enhance their ability to adapt to Moiré patterns and compression artifacts, ultimately improving detection accuracy. This improvement suggests that models benefit from being updated to handle new types of distortions or patterns, which may not have been fully accounted for in the original training process.

**IMAGE DISTORTION ATTACKS.** We evaluated the impact of Gaussian blurring and sharpening on deepfake detection by applying these techniques to the original datasets. Gaussian blurring, implemented with OpenCV's GaussianBlur function [82], smooths images by reducing fine details, while sharpening, using a high-pass filter via filter2D, enhances edges [83]. This systematic approach ensures consistent application, allowing direct comparison of detection performance. In Appendix Table 10, we present AUC scores before and after applying these transformations.

**MITIGATION STRATEGIES – PERFORMANCE AFTER DEMOIRÉING.** The top-performing deepfake detectors across all demoiréing techniques were CCViT [58], CADDM [67], and Rossler C23 [28], consistently ranking 1st, 2nd, and 3rd, respectively (see Appendix Table 11 for detailed results). CCViT achieved the highest average score of 79.2%, maintaining superior performance across original, Moiré-affected, and demoiréd images. Among demoiréing methods, ESDNet [80], trained on the FHDMi dataset, exhibited the lowest performance loss, indicating its effectiveness in mitigating Moiré-induced degradation. Conversely, DDA [81] demonstrated the highest performance loss, likely due to its optimization for mobile devices, which compromises its detection capabilities compared to other techniques. A significant finding from this experiment was

Figure 5: Moiré vs. Demoiréd

that while demoiréing methods effectively removed most Moiré patterns from the images (see Figure 5), they also eliminated certain deepfake artifacts that detectors rely on for classification. As a result, performance on demoiréd images dropped more than on images with Moiré patterns. Specifically, Moiré patterns caused an average performance drop of 10.1 percentage points for detectors. In contrast, demoiréd images resulted in an average drop of 14.7 percentage points. This underscores the need for advanced mitigation strategies to address Moiré patterns without inadvertently removing critical deepfake artifacts, ensuring robust detection performance. We conducted additional experiments by processing Moiré videos using VD-Moiré [84] and FPANet [85] demoiréing methods are outlined in Table 12 and denoising and deblurring from NAFNet [86], detailed results from these experiments are provided in (see Appendix Table 13 and Table 14).

Table 9: Overview of detectors with Fine-Tune and Retrain.

| DETECTORS | FINE-TUNE | | | | RETRAIN | | | |
|---|---|---|---|---|---|---|---|---|
| | OG | CMPA | SMPA-MA | SMPA-SPS | OG | CMPA | SMPA-MA | SMPA-SPS |
| Rossler C23 | 77.0 | 80.6 | 94.4 | 81.1 | 87.9 | 84.7 | **94.9** | 79.5 |
| MAT | **94.5** | **85.4** | 70.3 | **95.6** | **97.9** | **89.0** | 71.3 | **96.5** |
| CADDM | 86.3 | 84.6 | **94.4** | 95.0 | 85.1 | 81.9 | 92.9 | 95.4 |

**MITIGATION STRATEGIES – PERFORMANCE AFTER FINE-TUNING AND RETRAINING.** For fine-tuning, we utilized pretrained weights derived from the original dataset, which were also employed to assess the model's performance on the same data. The test dataset for fine-tuning and retraining comprises original data, captured Moiré data, and synthetic Moiré data. In the case of MAT, performance after retraining exhibited an improved score (see Table 9). However, for CADDM, fine-tuning demonstrated superior performance compared to retraining.

**ADDITIONAL ANALYSIS OF MOIRÉ IMPACT AND MITIGATION.** We evaluate eight image detectors on original datasets (CelebDF, DFD, DFDC, FF++, and UADFV) and under the most severe Moiré distortion (LED screen) with multiple variations (light on/off, iPhone 13/Samsung S22+). We also assess the performance after demoiréing, denoising, and deblurring effects. The corresponding ROC curves are presented in (see appendix Figure 15—Figure 25), showing varying performance on image detectors and random guess prediction when impacted by Moiré patterns. Furthermore, our investigation extends to evaluating the impact of Moiré patterns on frequency analysis, Appendix Figure 26,

and deepfake generative models, with results provided in (see Appendix Figure 27 and Figure 28), with non-GAN and GAN showing distinct frequency patterns.

**REMARKS.** These results demonstrate that just preprocessing methods (e.g., demoiréing) are insufficient to address the challenge posed by deepfake videos containing Moiré patterns or other artifacts. This highlights the need for more robust detection models capable of handling such distortions. In this context, our DMF dataset provides a valuable addition to public deepfake datasets for training these detectors.

# 6    Discussion

**Challenges in Data Collection.** Capturing Moiré patterns in real-world conditions required careful consideration of screen types, lighting variations, angles, and smartphone camera differences. Our dataset comprises 12,832 videos spanning 35.64 hours, sourced from CelebDF, DFD, DFDC, FF++, and UADFV, ensuring diverse representation. Differences in screen pixel structures influenced the intensity of Moiré artifacts. Additionally, smartphone cameras introduced variability in artifact appearance, further complicating the data collection process. These challenges highlight the complexity of generating a dataset that accurately represents Moiré-induced distortions in deepfake detection.

**Limitation and Future work.** While we acknowledge that real-world Moiré-inducing conditions span a wide range of factors, including variations in camera and display hardware, and dynamic motion, this work focuses on analyzing the impact of Moiré patterns on deepfake detection. Our experimental setup was intentionally designed to control these variables in a reproducible environment, enabling a focused investigation of Moiré-related effects. Broader scenarios involving diverse hardware configurations, motion artifacts, and platform-specific filters (e.g., beautification or AR effects on apps like TikTok and Instagram) remain essential directions for future work.

# 7    Conclusion

In this paper, we investigated the impact of Moiré patterns on deepfake detection, exposing a significant vulnerability in current methods. Our experiments showed that both Authentic and Synthetic Moiré patterns can degrade detector performance, reducing accuracy by up to 25.4%. This issue is further exacerbated by compression artifacts, where the combined effect leads to even greater performance deterioration. These findings highlight that existing models, often designed for clean, high-quality inputs, struggle with real-world artifacts introduced by screen captures and digital processing. While demoiréing techniques can mitigate these distortions, they may also inadvertently weaken detection performance. This underscores the need for more resilient deepfake detection systems capable of handling practical distortions like Moiré patterns and compression without significant accuracy loss.

**SOCIAL IMPACT.** Our work highlights the need for advanced deepfake detection to mitigate real-world artifacts. The dataset we share contains the real and deepfake videos captured with different mobile devices. The package also contains detailed documentation with all relevant metadata specified to users. We recommend using DMF as a training dataset to enhance detector robustness, aiding efforts to curb the spread of malicious deepfakes. To promote responsible, impactful use of the DMF dataset and to discourage misuse aimed at bypassing detectors, we provide access through a DOI-based request system. This process enhances security and ensures the dataset is used strictly for legitimate academic research.

# Acknowledgement

This work was partly supported by Institute for Information & communication Technology Planning & evaluation (IITP) grants funded by the Korean government MSIT: (RS-2022-II221199, RS-2022-II220688, RS-2019-II190421, RS-2023-00230337, RS-2024-00437849, RS-2021-II212068, and RS-2025-02263841). Also, this work was supported by the Cyber Investigation Support Technology Development Program (No.RS-2025-02304983) of the Korea Institute of Police Technology (KIPoT), funded by the Korean National Police Agency. Lastly, this work was supported by the National Research Foundation of Korea (NRF) grant funded by the Korea government (MSIT) (No. RS-2024-00356293).

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
