# Through the Lens: Benchmarking Deepfake Detectors Against Moiré-Induced Distortions

**Razaib Tariq**[*]      **Minji Heo**[*]      **Simon S. Woo**[†]                **Shahroz Tariq**[†]

Sungkyunkwan University, South Korea                CSIRO's Data61, Australia

{razaibtariq,minji.h0224,swoo}@g.skku.edu   shahroz.tariq@data61.csiro.au

[*]Equal contribution. [†]Corresponding author.

# Appendix

**Table of Contents**

## A  Real World Examples of Moiré Pattern

Deepfake videos are captured on the television screen using a smartphone device and distributed to different social networking services. The Figure 6 showcases how a novice tries to capture a deepfake when the platform will not allow the user to download it.

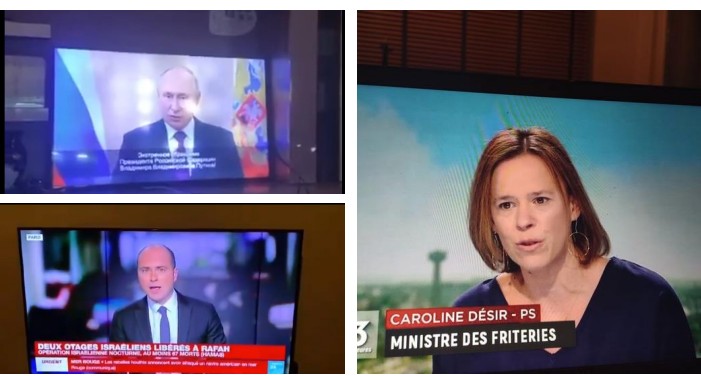

Figure 6: **The examples on both sides show video on Live Broadcast captured by a smartphone camera introducing the Moiré Pattern.**

## B  Visualization of Dataset Samples

Visualized examples in this section thoroughly compare the original images and those with Moiré patterns acquired on two separate monitors: LG (LED) and BenQ (LED), respectively, where the BenQ monitor tends to exhibit more pronounced Moiré patterns. The comparison involves displaying images before and after the implementation of ESDNet on different pre-trained weights, UHDM, and FHDMi for demoiréing, demonstrating how each method reduces Moiré patterns, and deblurring and denoising methods illustrate their efficacy in lowering Moiré patterns. This enables an evaluation of the effectiveness of each strategy on various displays.

### B.1  Examples of Original vs. Moiré Pattern

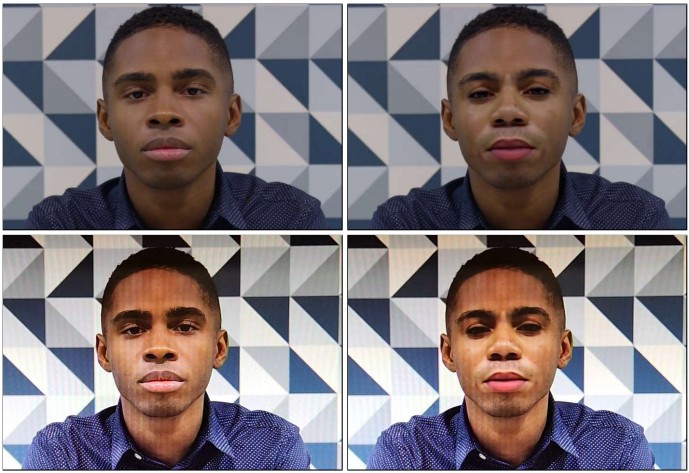

Figure 7: **Original vs. Moiré Pattern. Top-left:** Real image without Moiré Pattern. **Top-right:** Deepfake without Moiré Pattern. **Bottom-left:** Real image with Moiré Pattern. **Bottom-right:** Deepfake with Moiré Pattern.

## B.2 Examples of Moiré Patterns in Different Settings

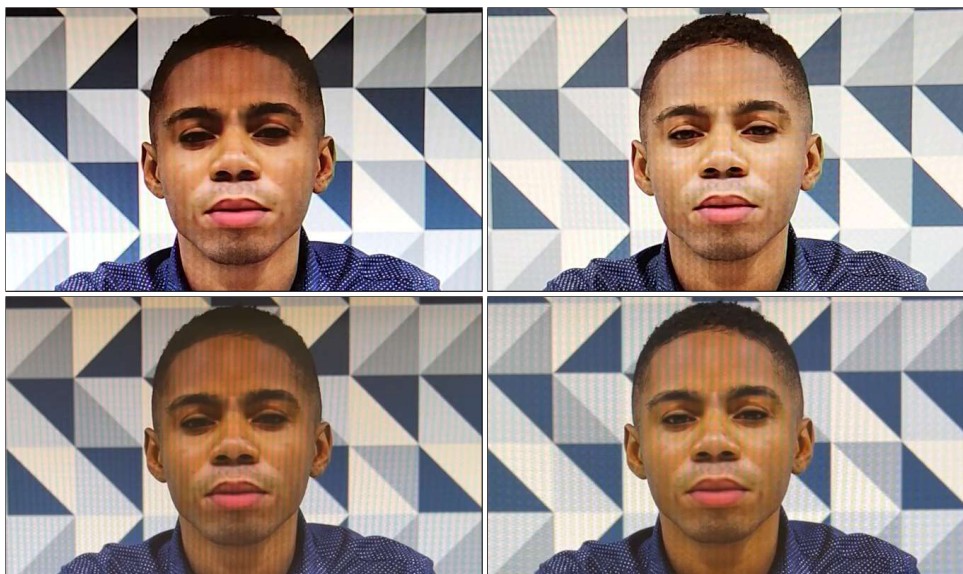

Figure 8: **Moiré Patterns on Different Monitors captured by Different Cameras.** **Top-left:** On LG Monitor captured by Samsung S22 Plus. **Top-right:** On BenQ Monitor captured by Samsung S22 Plus. **Bottom-left:** On LG Monitor captured by iPhone 13. **Bottom-right:** On BenQ Monitor captured by iPhone 13.

## B.3 Examples of Demoiréing using ESDNet (UHDM) Methods

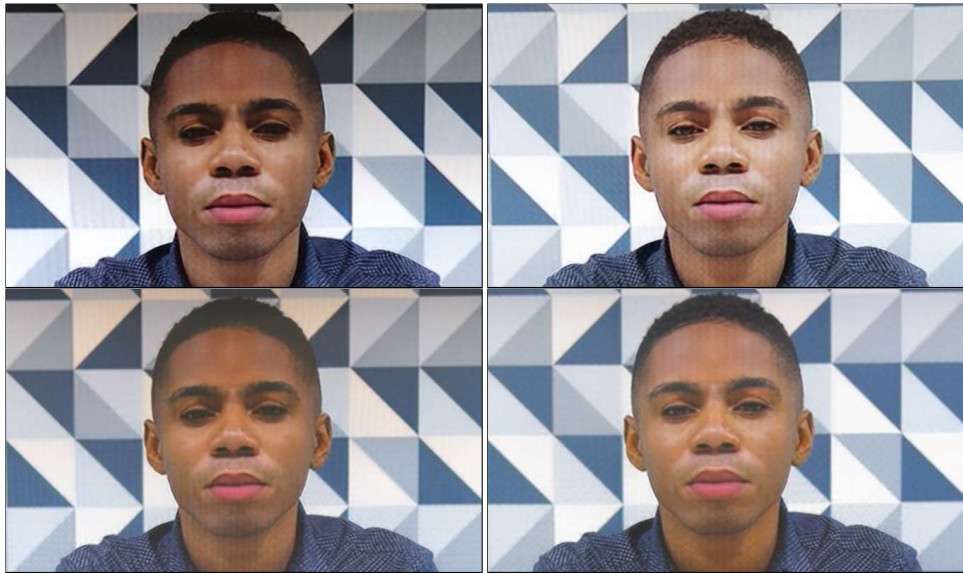

Figure 9: **Demoiré using the ESDNet (UHDM) Method. Top-left:** On LG Monitor captured by Samsung S22 Plus. **Top-right:** On BenQ Monitor captured by Samsung S22 Plus. **Bottom-left:** On LG Monitor captured by iPhone 13. **Bottom-right:** On BenQ Monitor captured by iPhone 13.

## B.4 Examples of Demoiréing using ESDNet (FHDMi) Methods

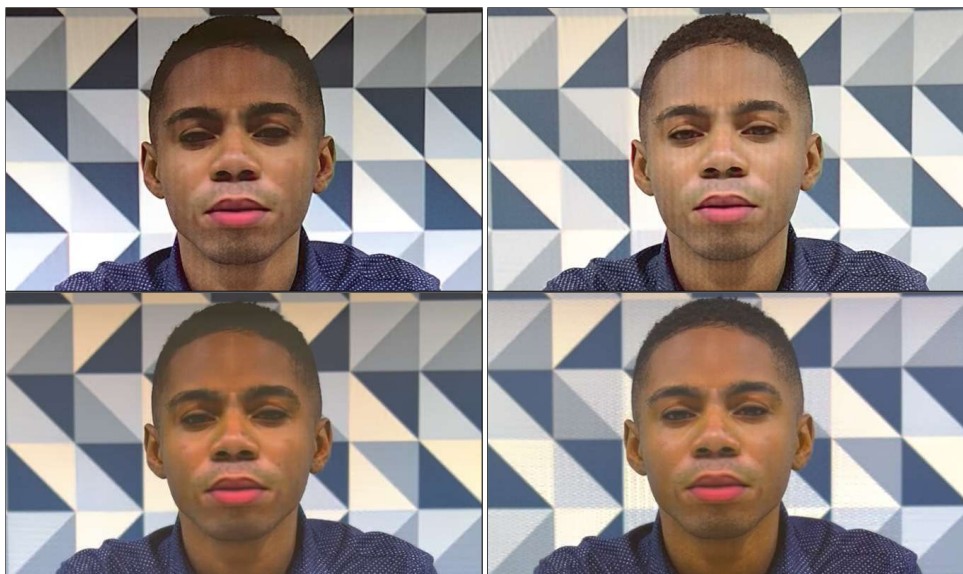

Figure 10: **DEMOIRÉ USING THE ESDNET (FHDMI) METHOD. Top-left:** On LG Monitor captured by Samsung S22 Plus. **Top-right:** On BenQ Monitor captured by Samsung S22 Plus. **Bottom-left:** On LG Monitor captured by iPhone 13. **Bottom-right:** On BenQ Monitor captured by iPhone 13.

## B.5 Examples of Deblurring and Denoising Methods.

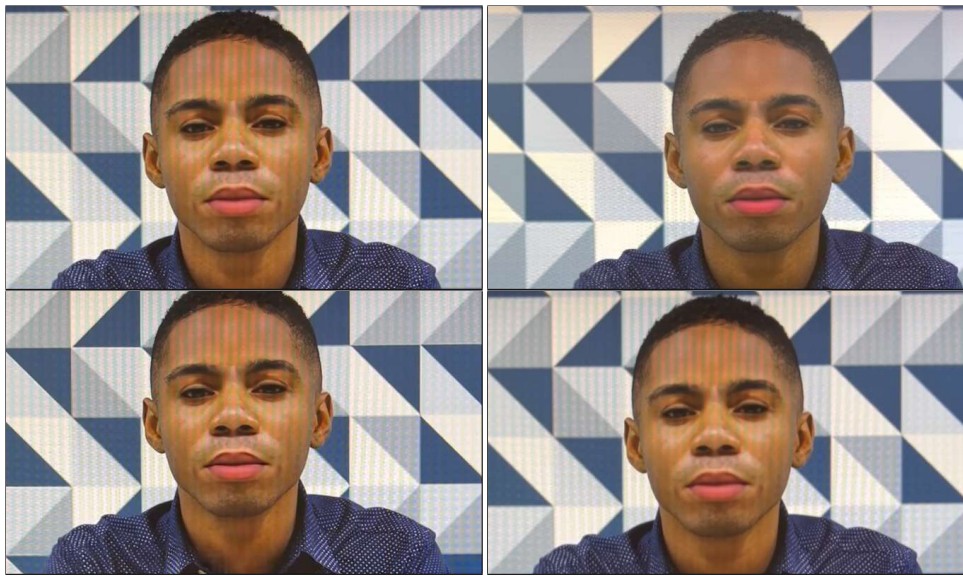

Figure 11: **DEMOIRÉING VS. DEBLURRING VS. DENOISING. Top-left:** Deepfake with Moiré Pattern. **Top-right:** Moiré image processed with the demoiréing (FHDMi) method. **Bottom-left:** Moiré image processed with the deblurring (GoPro-64) method. **Bottom-right:** Moiré image processed with the Denoising (SSID-64) method.

## B.6 Comparison of Original, Moiré Pattern, and Demoiréd Images.

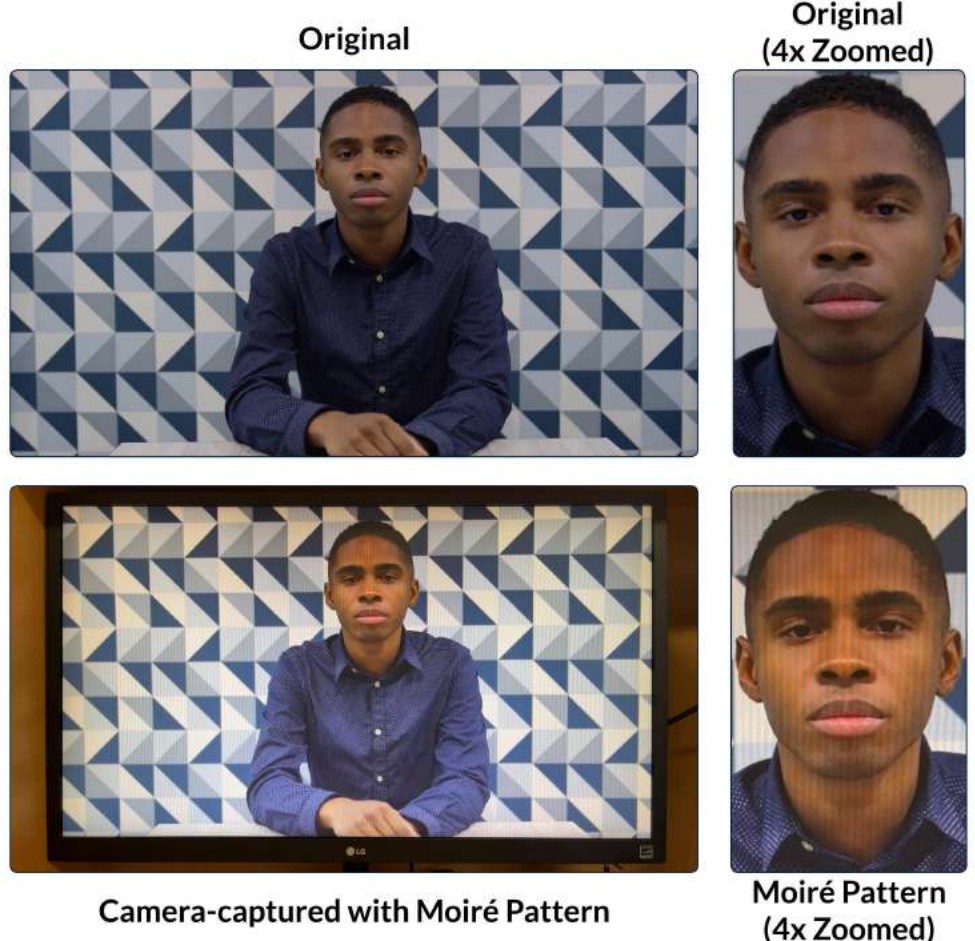

Figure 12: **ORIGINAL VS. MOIRÉ PATTERN IMAGES:** The top row displays original images without Moiré artifacts, with the leftmost image showing a full-frame original and the rightmost a 4× zoomed-in view. The bottom row illustrates how smartphone capture on a monitor introduces Moiré patterns, with the leftmost image showing the full-frame effect and the rightmost highlighting its distortions.

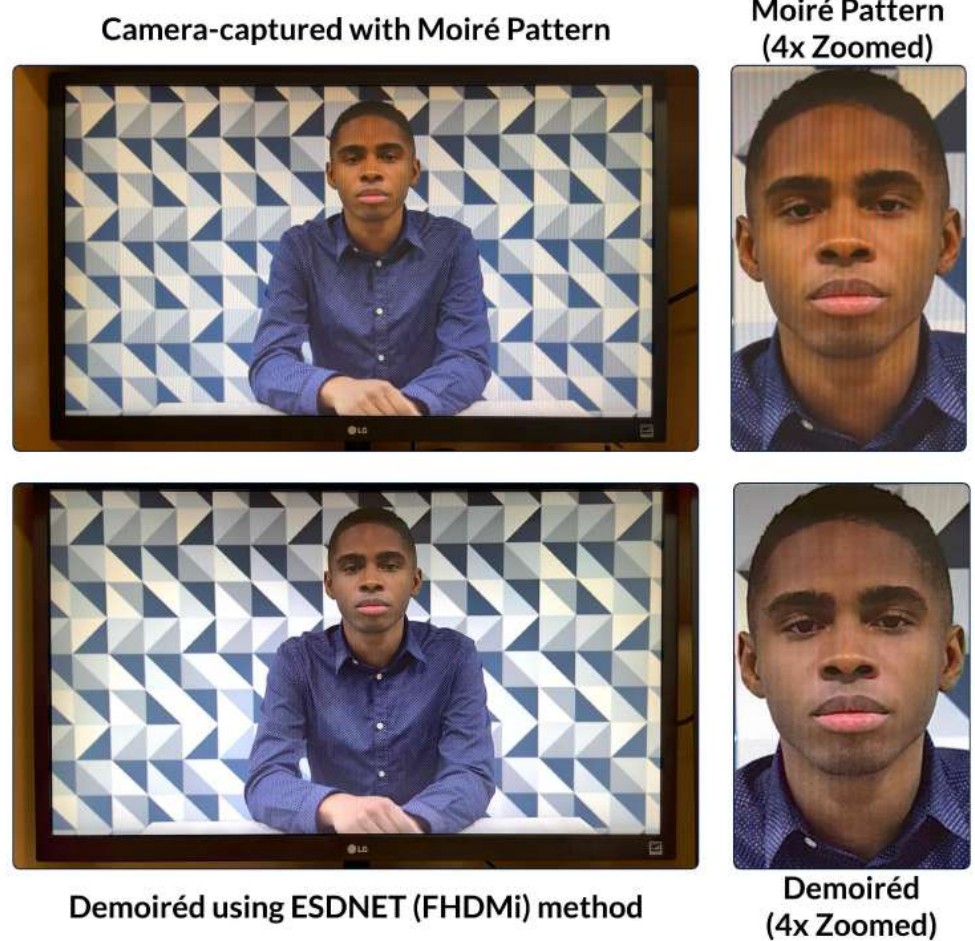

Figure 13: **Moiré vs. Demoiréd Images:** The top row shows smartphone-captured images with Moiré artifacts, with the leftmost image presenting the full-frame pattern and the rightmost a 4× zoomed-in view. The bottom row demonstrates the effect of ESDNet (FHDMi) in removing Moiré patterns, with the leftmost image showing the demoiréed result and the rightmost a zoomed-in comparison.

## C  Impact of Lighting Conditions on Moiré-Captured Data

Lighting conditions can influence how Moiré patterns appear in captured videos, potentially affecting the performance of deepfake detection models. To evaluate this, we compared detection accuracy under two ambient lighting setups: lights on and lights off. As shown in Figure 14, most detection models exhibited minimal variation in performance across the two conditions. Interestingly, models such as MAT, CCViT, and CADDM showed slightly higher scores when the lights were on. This may be attributed to increased ambient reflections and contrast enhancement, which can amplify the visibility of Moiré artifacts and make them easier for detectors to exploit. However, the overall difference was relatively small, suggesting that most models are robust to moderate changes in lighting during video capture. These results indicate that while lighting does have a minor effect on performance, the core challenge remains the presence of Moiré artifacts themselves rather than illumination conditions alone.

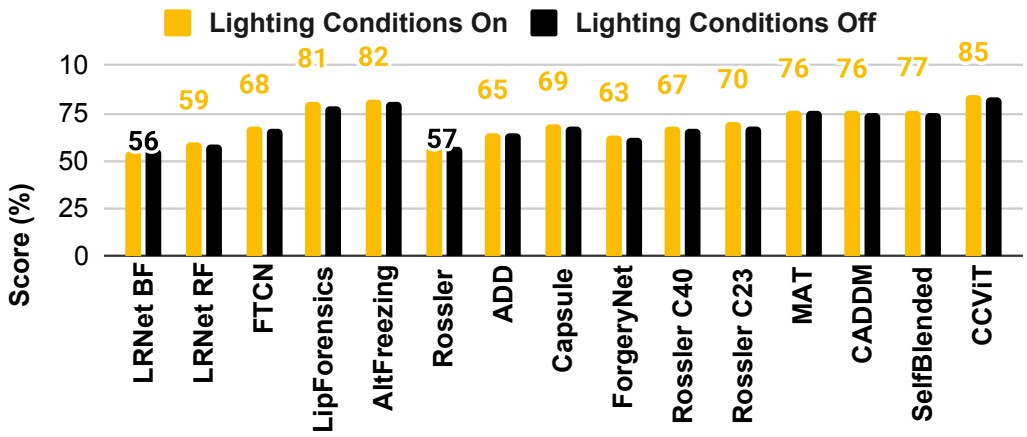

Figure 14: **Lighting Conditions:** Performance scores were similar across both conditions, though the 'lights on' (yellow) setting showed a slight improvement.**Lightning Conditions vs Performance for Each Detection Method.**

## D  Other Type of Distortions

Table 10 shows the AUC performance of Rossler C23 and MAT detectors under Gaussian blurring and sharpening. Both detectors experienced performance degradation as the blur kernel size increased, indicating vulnerability to low-pass filtering. Notably, MAT showed a sharper decline under severe blurring (15×15 kernel), suggesting greater sensitivity to the removal of fine-grained features. Conversely, sharpening generally improved detection performance, particularly for MAT, which achieved its highest AUC (98.2%) with a sharpening kernel value of 3. This suggests that enhancement of local edges and high-frequency details can accentuate manipulation artifacts, benefiting attention-based detectors. These results highlight the asymmetric effects of low-level image transformations on deepfake detectors and the importance of evaluating robustness under diverse perturbation types.

Table 10: **Performance Under Gaussian Blurring and Sharpening:** AUC scores of Rossler C23 and MAT detectors on original, blurred, and sharpened images.

| Detectors | Original | Gaussian Blurring (Kernel Size) | | Sharpening (Kernel Value) | |
|---|---|---|---|---|---|
| | | 7×7 | 15×15 | 3 | 5 |
| Rossler C23 | 86.5 | **74.5** | **67.5** | 88.5 | 76.7 |
| MAT | 87.0 | 71.3 | 57.7 | **98.2** | **88.6** |

## E Image Demoiréing Methods

We assessed the impact of applying four state-of-the-art demoiréing methods DMCNN, MBCNN, ESDNet (two variants), and DDA on deepfake detection performance using the DMF dataset. As these techniques are primarily designed for image-based restoration, evaluations were conducted using image-based detectors. To extend the analysis to video-level settings, LipForensic was additionally employed due to its capability to operate on both image and video modalities. The results in the following table reflect detection performance after demoiréing.

Table 11: **DIFFERENT DEMOIRÉING METHODS:** We tested four state-of-the-art methods, finding up to a 16% performance decline. This may be due to Demoiréing removing key deepfake artifacts needed for detection. Here, OG denotes Original.

| DETECTORS | AUC ON OG | AUC ON MOIRÉ | DEMOIRÉING METHODS PERFORMANCE | | | | | |
|---|---|---|---|---|---|---|---|---|
| | | | ESDNet (FHDMi) | ESDNet (UHDM) | MBCNN | DMCNN | DDA | Average |
| Rossler | 67.7 | 58.5 | 58.1 | 53.7 | 55.9 | 57.1 | 54.4 | 55.8 |
| ADD | 69.7 | 67.6 | 68.5 | 65.5 | 66.1 | 65.7 | 64.8 | 66.1 |
| Capsule | 71.3 | 70.4 | 69.4 | 60.7 | 60.5 | 62.0 | 59.9 | 62.5 |
| ForgeryNet | 76.9 | 64.3 | 64.4 | 60.4 | 54.6 | 61.3 | 51.2 | 58.4 |
| Rossler C40 | 77.0 | 68.2 | 69.1 | 66.6 | 64.2 | 66.5 | 63.3 | 65.9 |
| Rossler C23 | 86.5 | 72.8 | 76.9 | 69.2 | 67.3 | 71.5 | 66.5 | 70.3 |
| MAT | 87.0 | 75.2 | 75.5 | 66.0 | 63.6 | 65.9 | 63.4 | 66.9 |
| CADDM | 87.1 | 78.5 | 79.5 | 73.4 | 72.7 | 75.0 | 72.3 | 74.6 |
| SelfBlended | 88.8 | 78.4 | 73.6 | 60.7 | 70.1 | 70.6 | 69.3 | 68.9 |
| LipForensics | 90.6 | 83.3 | 67.2 | 66.1 | 66.1 | 71.6 | 65.9 | 67.3 |
| CCViT | **95.0** | **85.8** | **84.5** | **75.3** | **76.2** | **82.2** | **77.9** | **79.2** |
| **Avg. AUC loss (DeMoiré vs. Moiré)** | | | -1.5↓ | -7.8↓ | -7.8↓ | 4.9↓ | 8.6↓ | 6.1↓ |
| **Avg. AUC loss (DeMoiré vs. OG)** | | | -10.1↓ | -16.4↓ | -16.4↓ | 13.5↓ | 17.2↓ | 14.7↓ |

## F Video Demoiréing Methods

In Table 12, we present the AUC performance of three video-based deepfake detectors AltFreezing, FTCN, and LipForensics on original (clean), Moiré-affected, and demoiréd videos processed using VD-Moiré[6] and FPANet[7]. Overall, the table highlights the negative impact of Moiré patterns on detection performance, with all models experiencing performance drops when tested on Moiré-affected videos. Notably, AltFreezing's AUC fell significantly from 100.0% to 84.4% and further decreased to 74.7% with VD-Moiré, though it partially recovered to 92.9 with FPANet. FTCN showed inconsistent results, dropping from 56.3% to 43.8% on Moiré videos and failing to improve meaningfully with FPANet (40.6%), though VD-Moiré slightly boosted its score to 68.8%. LipForensics demonstrated the most resilience, with a minor drop from 100.0% to 87.5% and recovering to 90.6% with both demoiréing methods. These results suggest that while some models, like LipForensics, benefit modestly from demoiréing, others remain sensitive to artifacts even after processing, indicating the limited generalizability of current demoiréing techniques across different detector architectures.

Table 12: **AUC PERFORMANCE OF VIDEO-BASED DETECTORS ON CLEAN, MOIRÉ, AND DEMOIRÉD VIDEOS**

| Detector | Original | Moiré Video | VD-Moiré (Demoiré) | FPANet (Demoiré) |
|---|---|---|---|---|
| AltFreezing | **100.0** | 84.4 | 74.7 | **92.9** |
| FTCN | 56.3 | 43.8 | 68.8 | 40.6 |
| LipForensics | **100.0** | **87.5** | **90.6** | **90.6** |

---

[6] https://github.com/CVMI-Lab/VideoDemoireing
[7] https://github.com/kuai-lab/nn24_FPANet

# G  Denoising and Deblurring Methods

We used the denoising and deblurring methods from the NAFNet[8]. The NAFNet models were trained on SSID and GoPro datasets, using widths of 32 and 64 to balance computational efficiency and accuracy. Experiments revealed that adjusting the lengths of specific components can enhance performance for tasks such as reducing image noise (SSID) and removing blurriness (GoPro). The utilization of a 32-bit width aims to optimize computing efficiency, while a 64-bit width seeks to achieve higher precision, enabling the model to deliver optimal results within the computational limitations.

## G.1  Evaluation of Denoising Method on Different Weights

Table 13: **DENOISING METHOD:** In the NAFNet (SSID) technique, we observed an unexpected decrease in performance of up to 19.5 percentage points due to the denoising process. This decline in detection accuracy is likely attributable to the denoising procedure. Furthermore, we found that the performance of the methods was less effective after denoising than after demoiréing.

| DETECTORS | AUC ON ORIGINAL | AUC ON MOIRÉ | DENOISING METHOD PERFORMANCE | | |
|---|---|---|---|---|---|
| | | | NAFNet (SSID-32) | NAFNet (SSID-64) | Average |
| Rossler | 67.7 | 58.5 | 60.71 | 59.62 | 60.16 |
| ADD | 69.7 | 67.6 | 65.93 | 68.96 | 67.44 |
| Capsule | 71.3 | 70.4 | 54.53 | 56.73 | 55.63 |
| ForgeryNet | 76.9 | 64.3 | 53.02 | 56.73 | 54.87 |
| Rossler C40 | 77.0 | 68.2 | 48.63 | 50.82 | 49.72 |
| Rossler C23 | 86.5 | 72.8 | 66.76 | 67.58 | 67.17 |
| MAT | 87.0 | 75.2 | 84.89 | 85.16 | 85.02 |
| CADDM | 87.1 | 78.5 | 61.26 | 66.76 | 64.01 |
| SelfBlended | 88.8 | 78.4 | 75.55 | 72.25 | 73.90 |
| LipForensics | 90.6 | 83.3 | 66.90 | 65.93 | 66.41 |
| CCViT | 95.0 | 85.8 | 79.95 | 76.92 | 78.43 |
| Avg. Performance loss (DeNoise vs. Moiré) | | | 7.7↓ | 6.9↓ | -7.3 |
| Avg. Performance loss (DeNoise vs. Original) | | | 16.3↓ | 15.4↓ | -15.9 |

## G.2  Evaluation of Deblurring Method on Different Weights

Table 14: **DEBLURRING METHOD:** We implemented the NAFNet (GoPro) technique for deblurring. Upon comparing the effectiveness of each method for demoiréing and denoising, we found that the deblurring technique exhibited the lowest performance of the two. The deblurring process led to an additional decrease in performance of up to 36.8 percentage points.

| DETECTORS | AUC ON ORIGINAL | AUC ON MOIRÉ | DEBLURRING METHOD PERFORMANCE | | |
|---|---|---|---|---|---|
| | | | NAFNet (GoPro-32) | NAFNet (GoPro-64) | Average |
| Rossler | 67.7 | 58.5 | 48.63 | 47.66 | 48.14 |
| ADD | 69.7 | 67.6 | 46.70 | 46.29 | 46.49 |
| Capsule | 71.3 | 70.4 | 52.06 | 50.14 | 51.10 |
| ForgeryNet | 76.9 | 64.3 | 46.43 | 46.98 | 46.70 |
| Rossler C40 | 77.0 | 68.2 | 45.33 | 44.23 | 44.78 |
| Rossler C23 | 86.5 | 72.8 | 60.03 | 55.08 | 57.55 |
| MAT | 87.0 | 75.2 | 77.75 | 74.31 | 76.03 |
| CADDM | 87.1 | 78.5 | 64.42 | 63.74 | 64.08 |
| SelfBlended | 88.8 | 78.4 | 59.34 | 60.44 | 59.89 |
| LipForensics | 90.6 | 83.3 | 69.09 | 61.26 | 65.17 |
| CCViT | 95.0 | 85.8 | 64.97 | 63.32 | 64.14 |
| Avg. Performance loss (DeBlur vs. Moiré) | | | 17.3↓ | 18.6↓ | -17.1 |
| Avg. Performance loss (DeBlur vs. Original) | | | 24.1↓ | 25.1↓ | -25.7 |

---

[8]https://github.com/megvii-research/NAFNet

# H   Performance on Original Dataset — ROC Curve

We conducted a comprehensive performance analysis of the original dataset, employing various methodologies on many datasets to evaluate the effectiveness of each method. This analysis aimed to assess the efficacy of four techniques moiré, demoiréing, denoising, and deblurring. Additionally, compare their performance to that of the original dataset.

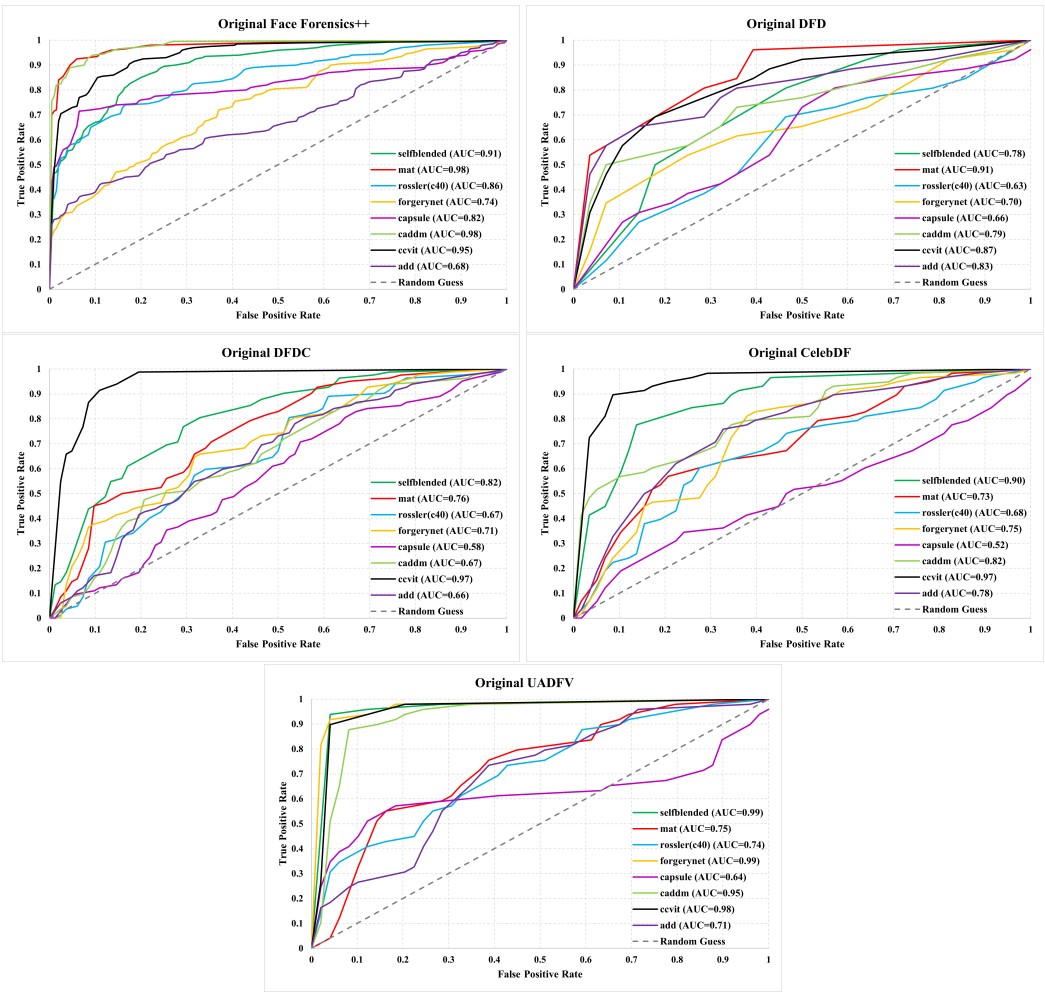

Figure 15: **PERFORMANCE ON ORIGINAL DATASETS.** The ROC AUC curves of different Deepfake Detectors on the Original deepfake datasets are unaffected by the Moiré-induced distortions. On FaceForensics++, MAT and CADDM models attain near-perfect AUCs of 0.98%, with CCViT performing well at 0.95%. For DFD, the MAT model leads with an AUC of 0.91%, followed by CCViT. In DFDC, CCViT again excels with an AUC of 0.97%, while Capsule has the lowest AUC of 0.58%. On CelebDF, it achieves an impressive AUC of 0.97%, while the Capsule model performs poorly with an AUC of 0.52%. For UADFV, SelfBlended and ForgeryNet models achieve AUCs of 0.99%, closely followed by CCViT at 0.98%. Capsule remains the least effective with an AUC of 0.64%. CCViT consistently ranks in the top 3 across all datasets, showcasing its reliable performance.

# I    Performance on DeepMoiréFake (DMF) Dataset — ROC Curve

For the performance evaluation on the DMF Dataset, we mainly used a dataset obtained from a BenQ monitor, which displays the most pronounced Moiré patterns. The BenQ monitor was chosen to ensure the dataset presented substantial challenges, allowing us to effectively assess and measure the efficiency of our approaches in demanding scenarios. The Moiré patterns produced by the BenQ monitor provided a solid basis for evaluating the effectiveness of various methods on each dataset in distinguishing real and fake videos.

## I.1    Camera: Samsung S22 Plus

### I.1.1    Lights Condition: ON

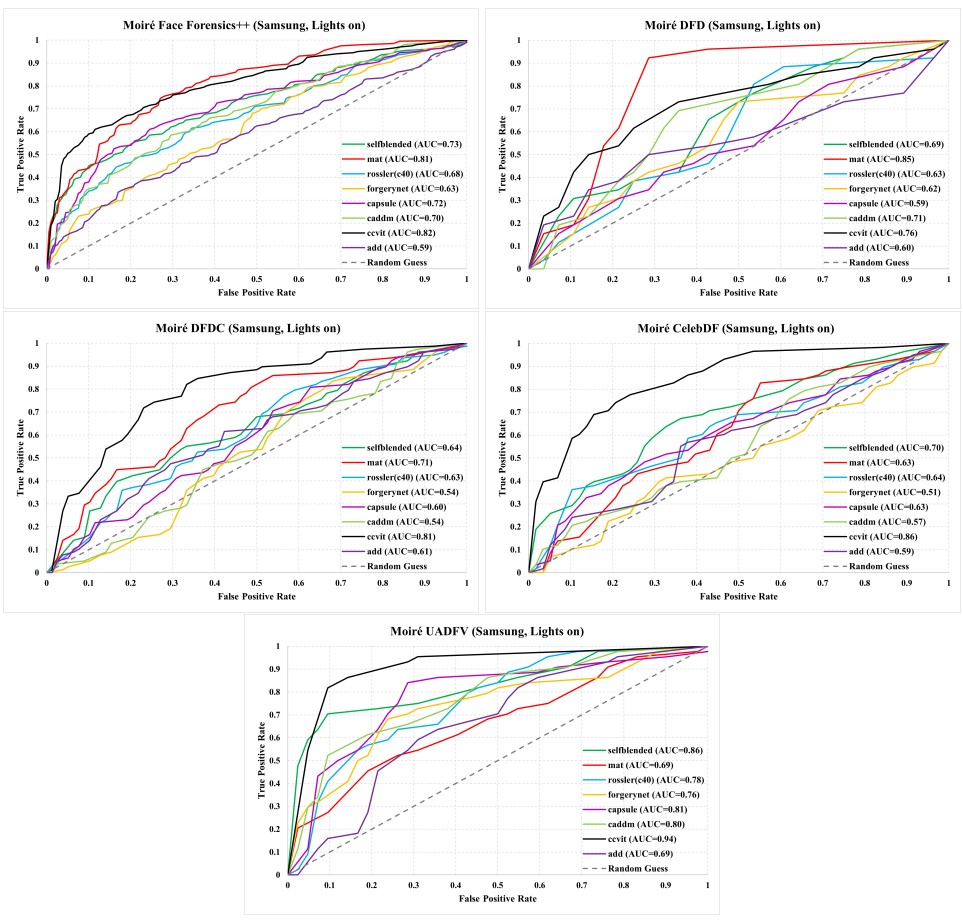

Figure 16: **Performance on Moiré pattern-induced Datasets on BenQ Monitor with Lights on.** The CCViT and MAT models generally perform best across different datasets. The performance of other models varies across datasets. On Moiré FaceForensics++, the CCViT and MAT models achieve AUC scores of 0.82% and 0.81%, respectively. In the Moiré DFD dataset, the MAT model leads with an AUC of 0.85%, followed by CCViT with 0.76%. For Moiré DFDC, CCViT scores 0.81%, while MAT achieves 0.71%. In Moiré CelebDF, CCViT leads with an AUC of 0.86%. In Moiré UADFV, CCViT excels with an AUC of 0.94%, followed by Capsule with 0.81%.

## I.1.2 Lights Condition: OFF

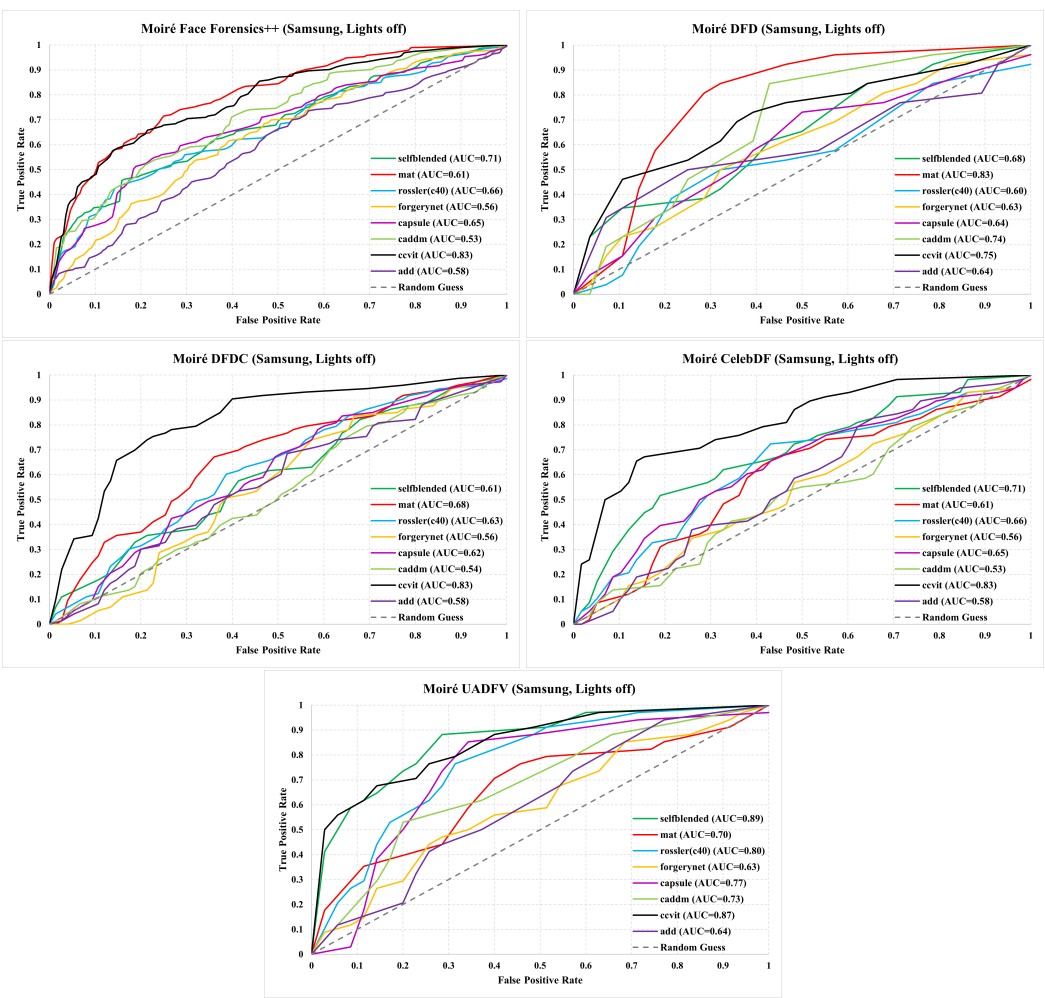

Figure 17: **PERFORMANCE ON MOIRÉ PATTERN-INDUCED DATASETS ON BENQ MONITOR WITH LIGHTS OFF.** For Moiré FaceForensics++, MAT leads with an AUC of 0.61%, closely followed by CCViT at 0.83%. In the Moiré DFD dataset, the MAT model leads with an AUC of 0.83%, followed by CCViT with 0.75%. In the Moiré DFDC dataset, CCViT performs best with an AUC of 0.83%, followed by the MAT model with 0.68%. For the Moiré CelebDF dataset, CCViT achieves the highest AUC of 0.83%, followed by SelfBlended at 0.71%. In the Moiré UADFV dataset, CCViT excels with an AUC of 0.87% and SelfBlended scores of 0.89%.

## I.2   Camera: iPhone 13

### I.2.1   Lights Condition: ON

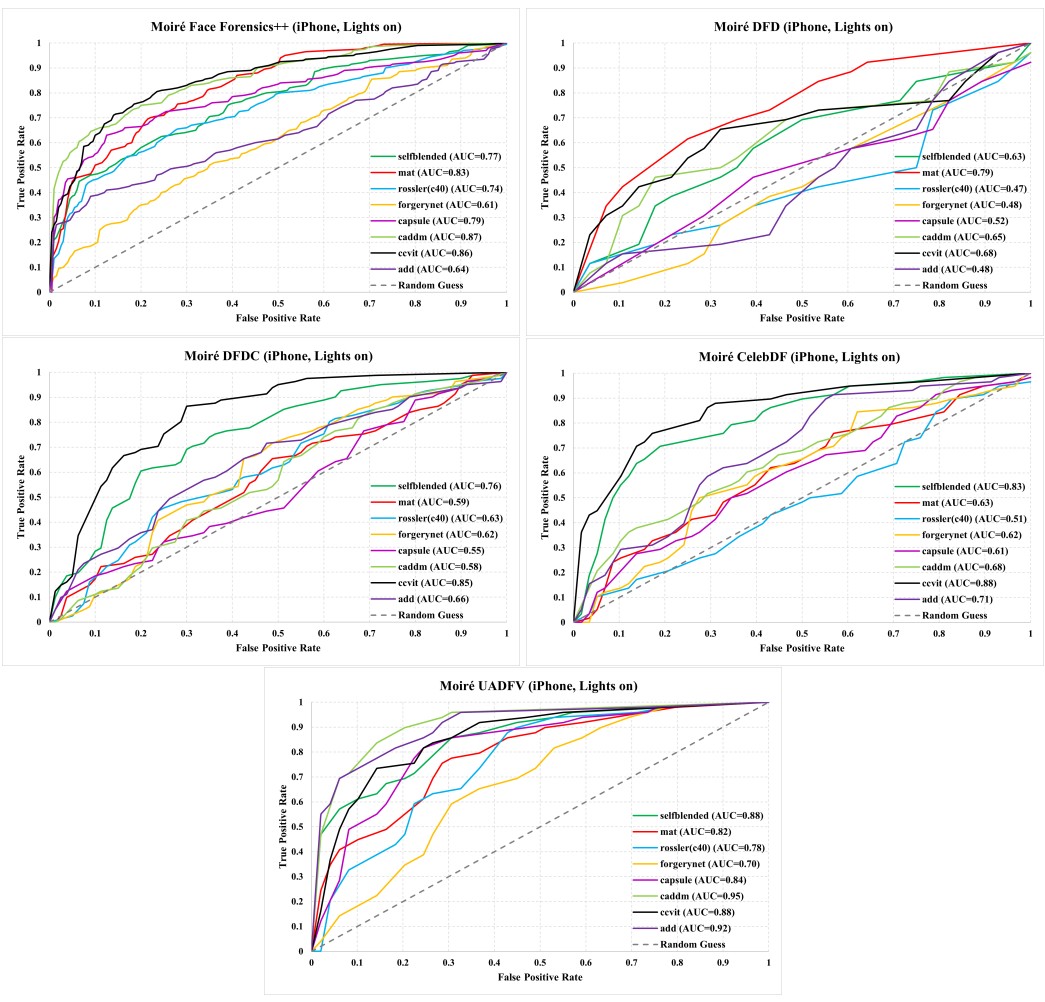

Figure 18: **Performance on Moiré pattern-induced Datasets on BenQ Monitor with Lights on.** The ROC curves display the performance of various deepfake detection methods on datasets captured with an iPhone in a light-on environment using a BenQ monitor, where moiré patterns were prominent. Across multiple datasets, the CCViT model consistently showed strong performance with AUCs of 0.86% in Moiré FaceForensics++, 0.85% in Moiré DFDC, and 0.88% in Moiré CelebDF. The MAT model also performed well, with AUCs of 0.83% in Moiré FaceForensics++ and 0.79% in Moiré DFD. Additionally, SelfBlended and CADDM models demonstrated notable performance, particularly in the Moiré UADFV dataset, achieving AUCs of 0.88% and 0.95%, respectively. CCViT and MAT models were the most reliable across various datasets under these conditions.

### I.2.2 Lights Condition: OFF

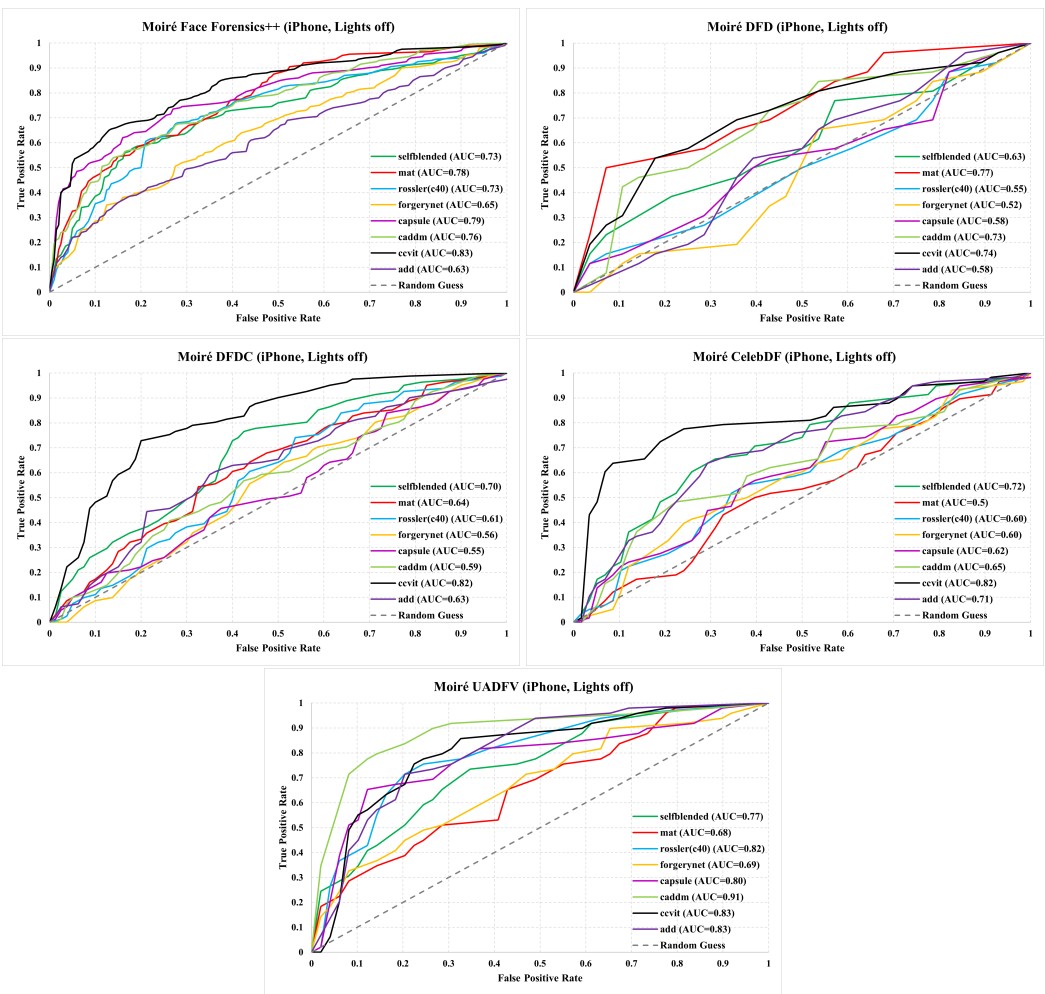

Figure 19: **PERFORMANCE ON MOIRÉ PATTERN-INDUCED DATASETS ON BENQ MONITOR WITH LIGHTS OFF.** CCViT consistently performs well across different datasets, achieving the highest AUCs of 0.83% in Moiré FaceForensics++ and 0.82% in both Moiré DFDC and CelebDF. MAT also shows strong performance with an AUC of 0.77% in Moiré DFD and 0.78% in Moiré FaceForensics++. Other notable performances include SelfBlended with an AUC of 0.72% in Moiré CelebDF and CADDM with the highest AUC of 0.91% in Moiré UADFV. Forgerynet generally shows lower performance across datasets. CCViT and MAT are the most reliable models for detecting deepfakes in these settings.

# J    Performance after Demoiréing on DMF Dataset — ROC Curve

To compare performance on demoiré datasets, we compare detection results on deepfake datasets captured with a BenQ monitor, where the moiré pattern was most pronounced. This comparison utilized the ESDNet (FHDMi) method, the best-performing demoiré technique, and the evaluation was carried out using ROC curves.

## J.1    Camera: Samsung S22 Plus

### J.1.1    Lights Condition: ON

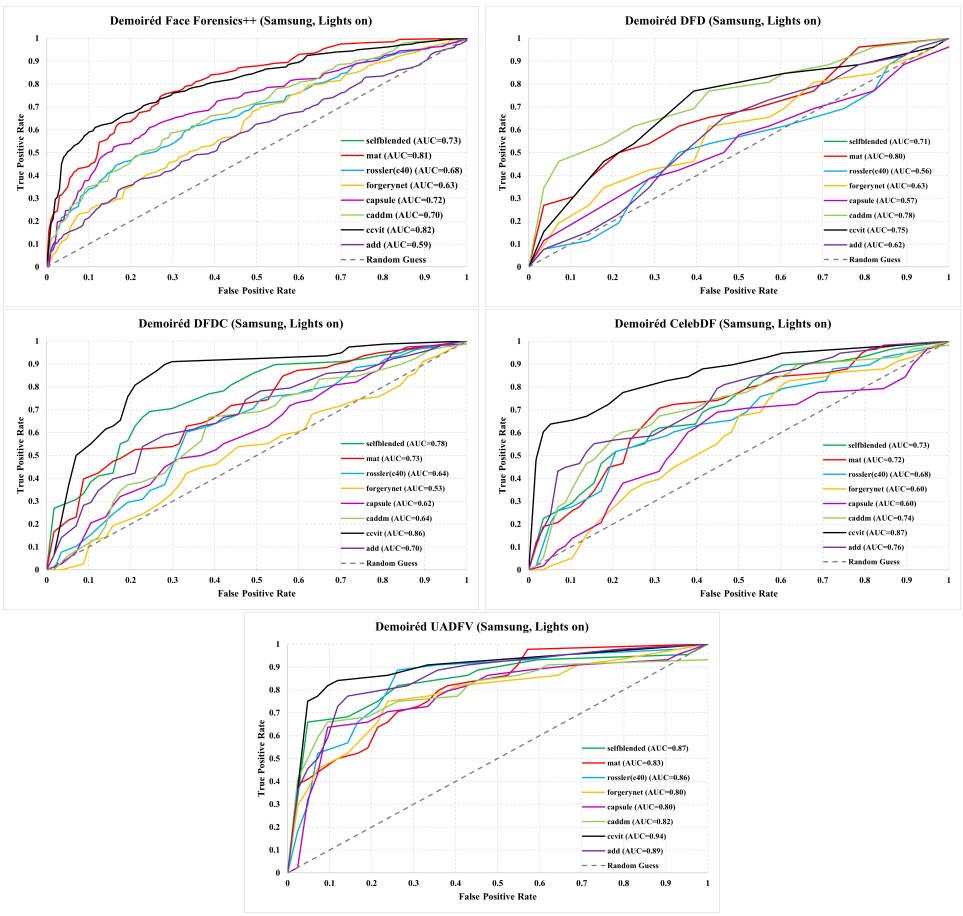

Figure 20: **Performance on Demoiré Datasets (BenQ Monitor, Lights On)** For the FaceForensics++ dataset, the CCViT model achieves the highest AUC of 0.82%, closely followed by the MAT model with 0.81%. MAT leads with an AUC of 0.80% in the DFD dataset, while CCViT follows with 0.75%. In the DFDC dataset, CCViT outperforms other models with an AUC of 0.86%. For the CelebDF dataset, CCViT again leads with an AUC of 0.87%. In the UADFV dataset, CCViT excels with an AUC of 0.94%, followed by the ADD model with 0.89%.

## J.1.2 Lights Condition: OFF

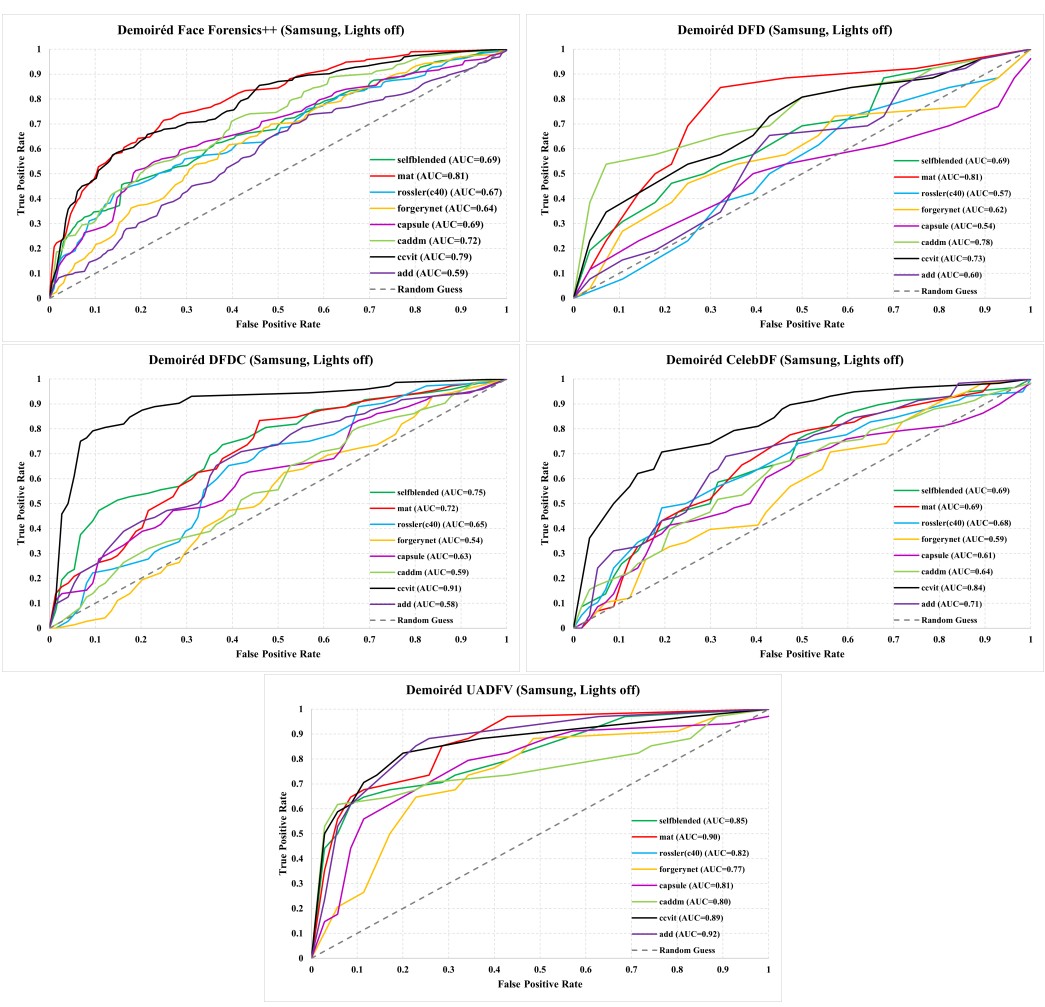

Figure 21: **PERFORMANCE ON DEMOIRÉ DATASETS (BENQ MONITOR, LIGHTS OFF)** In the FaceForensics++ dataset, MAT and CCViT demonstrate the highest AUC scores of 0.81% and 0.79%, respectively, indicating strong detection capabilities. MAT and CCViT again lead with AUC scores of 0.81% and 0.73% for the DFD dataset. In the DFDC dataset, CCViT performs best with an AUC of 0.91%, significantly outperforming other methods. The CelebDF dataset shows CCViT as the top performer with an AUC of 0.84%, followed by ADD at 0.71%. For the UADFV dataset, ADD achieves the highest AUC of 0.92%, with MAT following at 0.90%. These results suggest that MAT and CCViT are consistently effective across various datasets.

## J.2   Camera: iPhone 13

### J.2.1   Lights Condition: ON

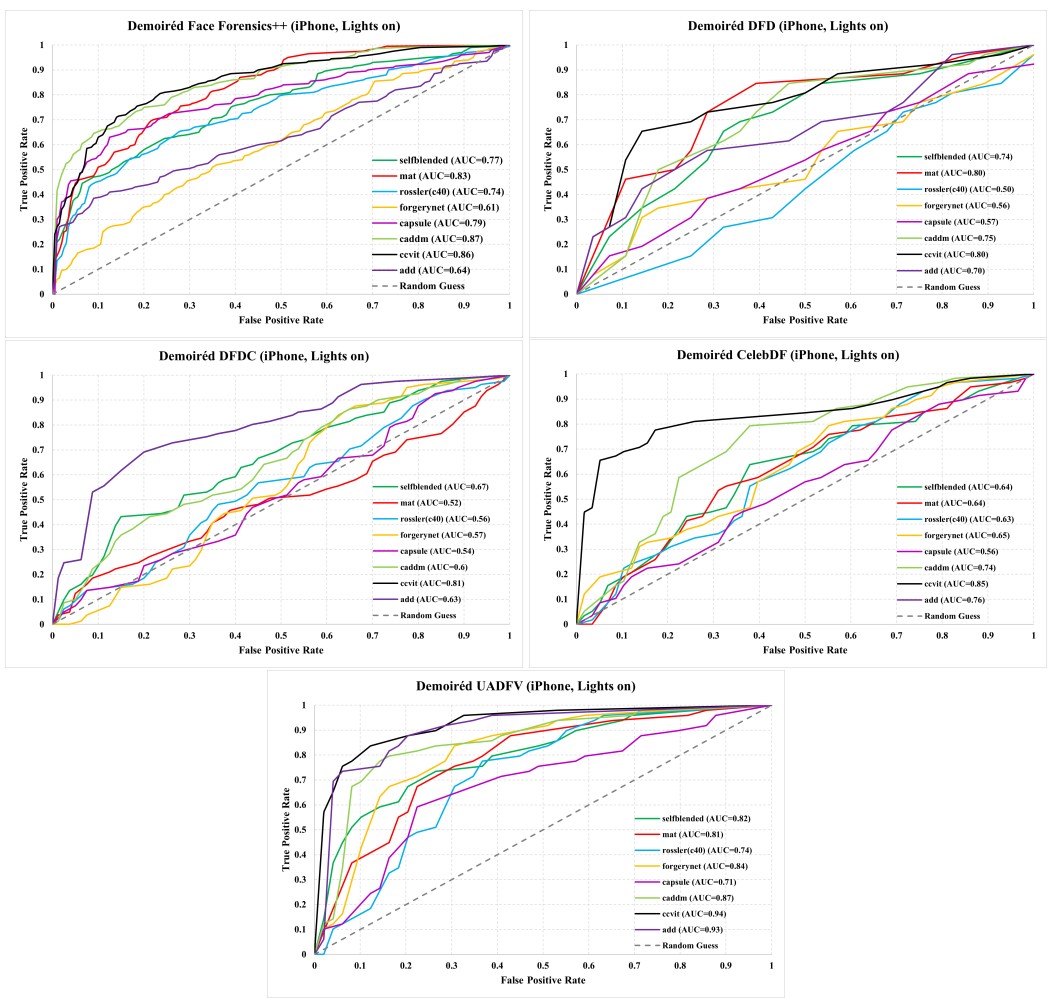

Figure 22: **PERFORMANCE ON DEMOIRÉ DATASETS (BENQ MONITOR, LIGHTS ON)** In the Face-Forensics++ dataset, the CADDM method achieved the highest AUC of 0.87%, followed by the CCViT method at 0.86%. The MAT method leads with an AUC of 0.80% for the DFD dataset, closely followed by CCViT with 0.80%. In the DFDC dataset, CCViT performs best with an AUC of 0.81%, with CADDM trailing at 0.60%. For the CelebDF dataset, CCViT also excels with an AUC of 0.85%, followed by the ADD method at 0.76%. Lastly, in the UADFV dataset, CCViT demonstrates superior performance with an AUC of 0.94%, while the ADD method achieves an AUC of 0.93%.

## J.2.2   Lights Condition: OFF

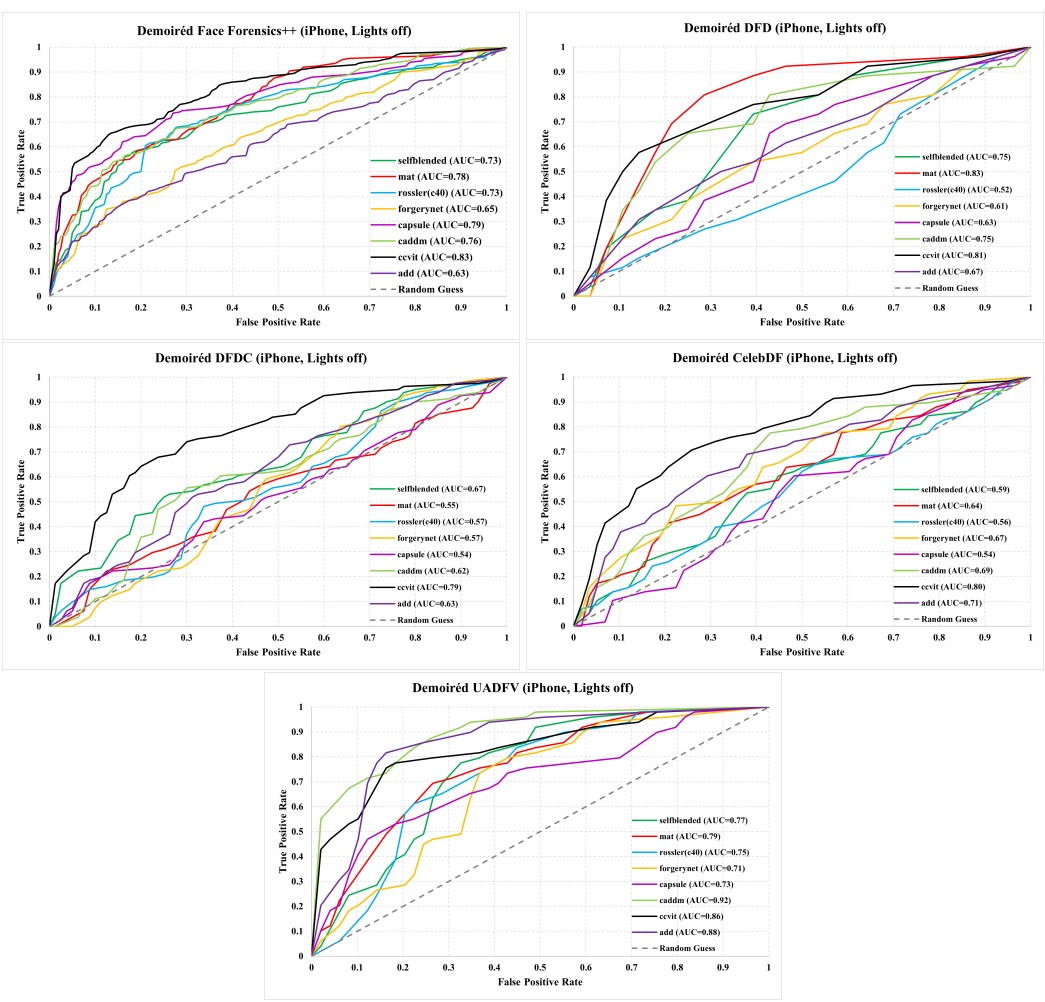

Figure 23: **PERFORMANCE ON DEMOIRÉ DATASETS (BENQ MONITOR, LIGHTS OFF)** The CCViT method consistently attained the highest AUC scores across various datasets, showcasing its superior performance. In detail, CCViT achieved AUC scores of 0.83% for FaceForensics++, 0.81% for DFD, 0.79% for DFDC, 0.80% for CelebDF, and 0.86% for UADFV datasets. The MAT and CADDM methods also exhibited strong performance, although their effectiveness varied depending on the dataset.

# K  Performance after Denoising on DMF Dataset — ROC Curve

## K.1  Camera: iPhone 13—Lights Condition: ON

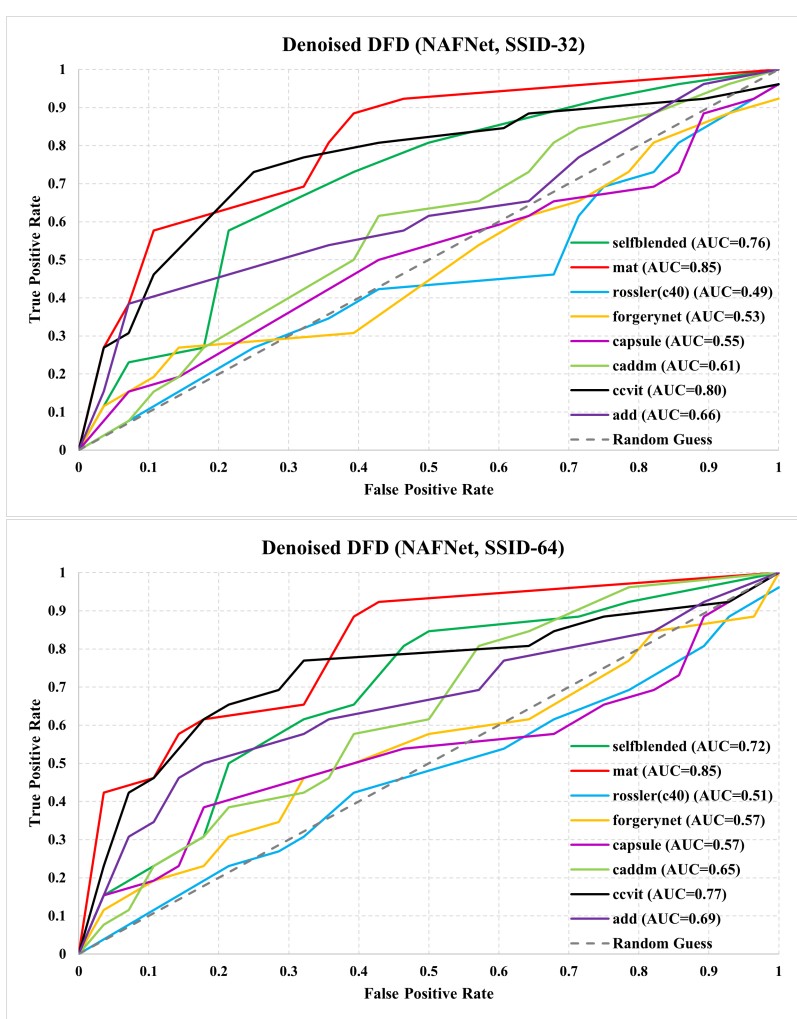

Figure 24: **PERFORMANCE ON DENOISE DATASETS (BENQ MONITOR, LIGHTS ON)** When tested on different weights of NAFNet (SSID-32/64) to remove the Moiré pattern, the MAT approach consistently achieved a high AUC of 0.85%. Based on this observation, it is evident that even with applying some denoise to the Moiré Pattern, MAT can keep its AUC in accurately predicting synthetic images on the DFD dataset. CCViT achieved the second-highest Area Under the Curve (AUC) score for both weights at 0.80% and 0.77%. Rossler et al. (C40) exhibited worse performance of 0.49% and 0.51% when subjected to the weight testing of NAFNet (SSID-32/64).

# L    Performance after Deblurring on DMF Dataset — ROC Curve

## L.1    Camera: iPhone 13—Lights Condition: ON

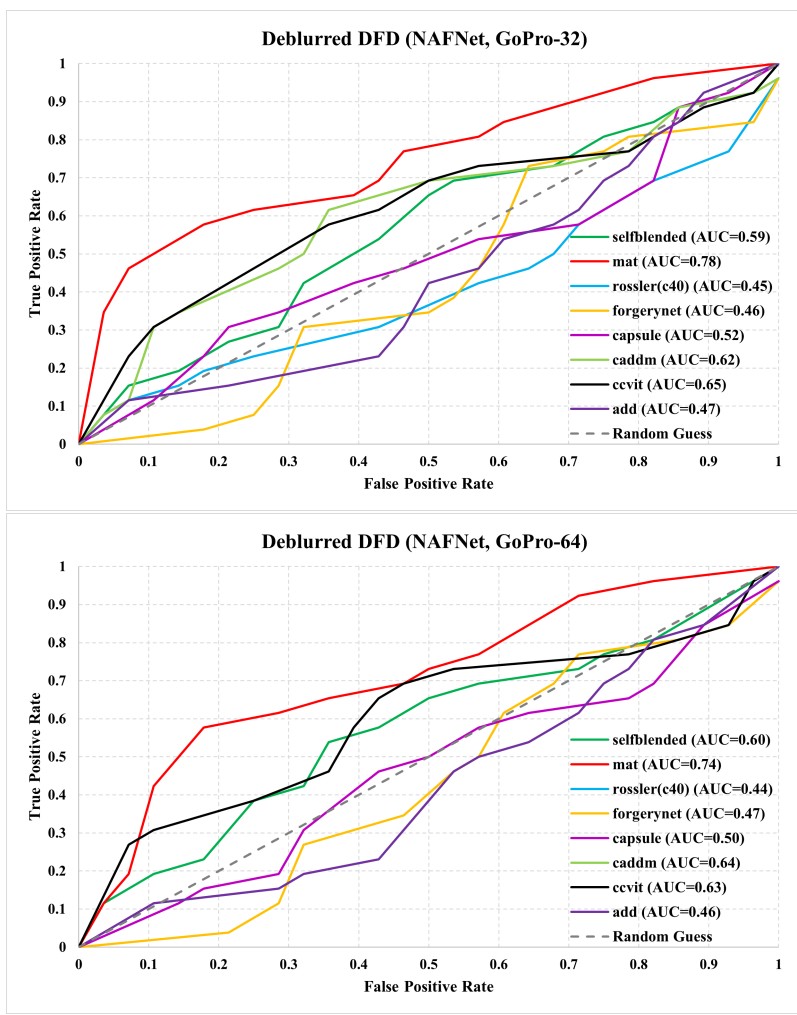

Figure 25: **PERFORMANCE ON DEBLUR DATASETS (BENQ MONITOR, LIGHTS ON)** When deblurring is used to eliminate the Moiré pattern, MAT demonstrates lower performance compared to denoising at 0.78% and 0.74%. However, the effectiveness of deblurring is noticeably reduced, suggesting that it also eliminates crucial elements that deepfake detectors rely on for predictions. The AUC of CCViT on NAFNet (GoPro-32) is the second highest at 0.65%, whereas CADDM achieves the highest AUC of 0.64% when tested with NAFNet (GoPro-64). ADD exhibited the lowest performance in both cases, with rates of 0.47% and 0.46%, respectively.

# M    Experimental Environment and Training Setup

We used TITAN RTX, RTX A5000, and RTX 3090 GPUs to preprocess and test image and video detectors and demoiréing methods. For both finetuning and retraining, we modified only the batch size (16) and number of epochs (10), while keeping the original optimizers, learning rates, and other hyperparameters as specified by the authors. Specifically, we used optimizer Adam with a learning rate of (1e-3) for Rossler, and AdamW for both MAT (1e-4) and CADDM (1e-3), ensuring consistency with the original training setups.

# N    Visual Analysis of Moiré Pattern on Frequency Spectrum

The figures in Figure 26 illustrate how Moiré patterns affect the frequency spectrum. Compared to the original spectrum, Moiré-affected real and fake samples exhibit noticeable distortions, disrupting the clear frequency structure. This degradation helps explain the drop in performance of both image- and video-based deepfake detectors. We also evaluated whether video demoiréing methods could mitigate this impact. As shown in Table 12, methods like LipForensics slightly improved after applying demoiréing. However, other methods, such as AltFreezing and FTCN, exhibited further accuracy decline, even after processing with FPANet, suggesting that these models still struggled to recover accurate predictions despite Moiré removal.

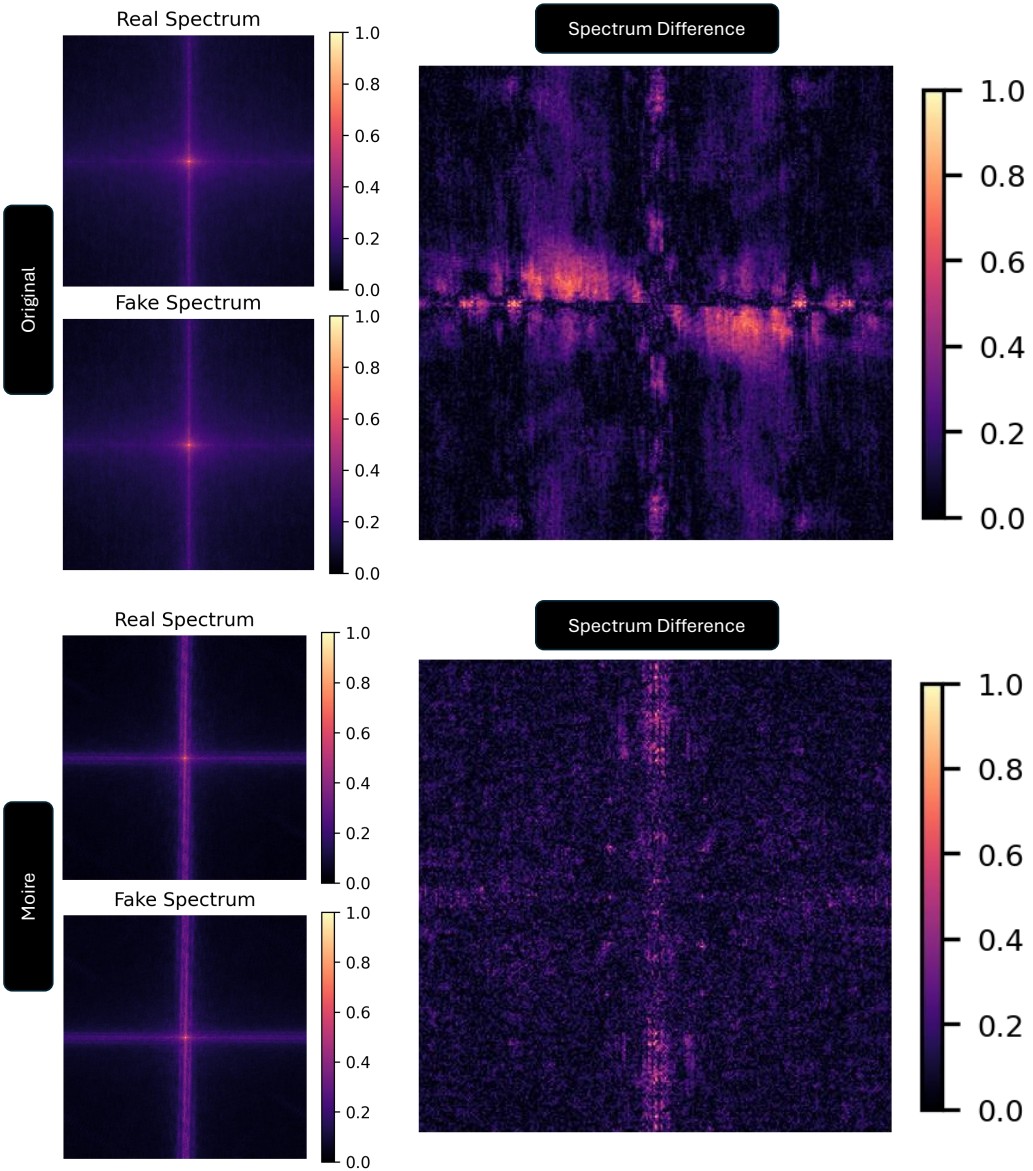

**Frequency spectra of Original and Moire pattern**

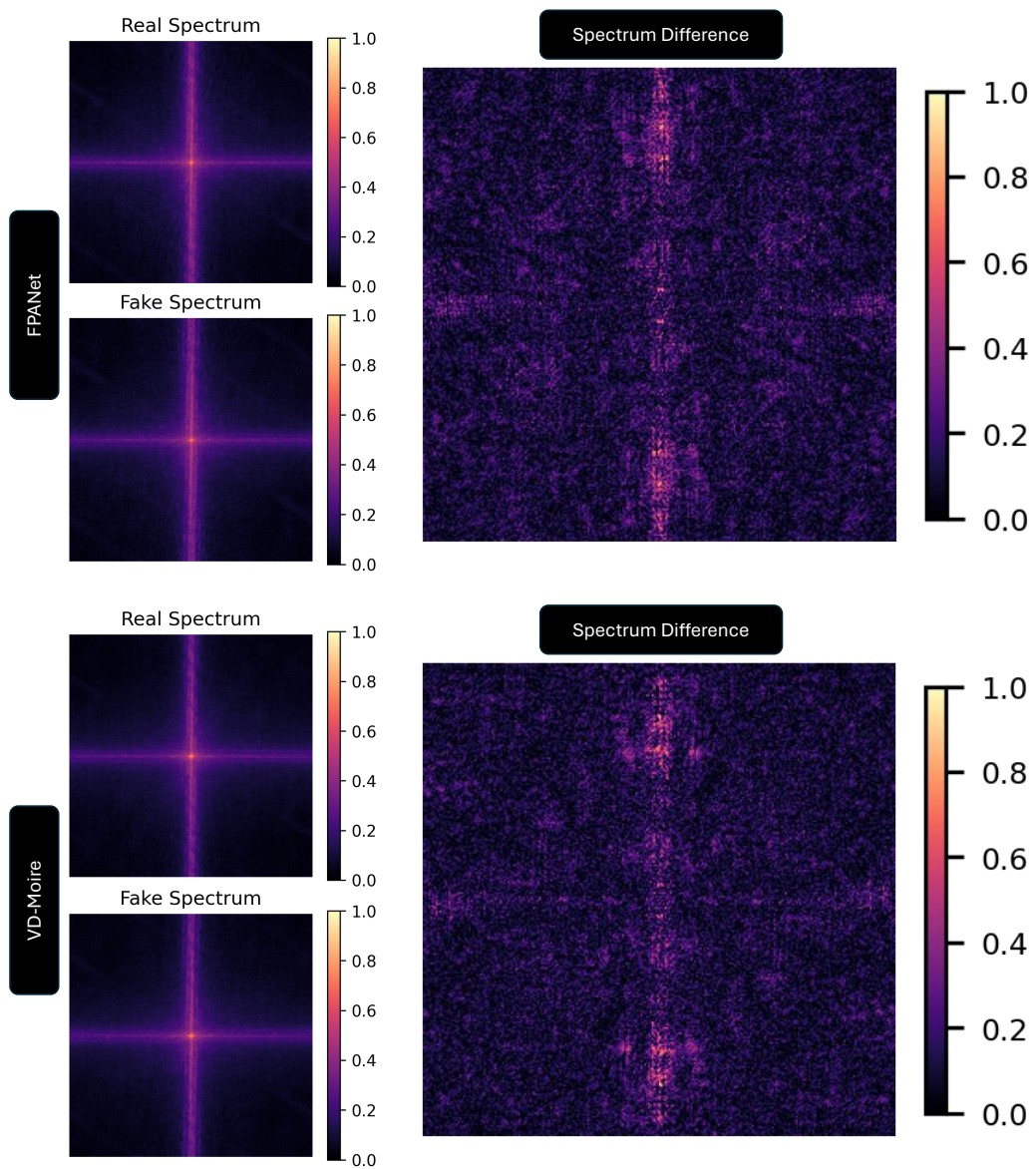

Figure 26: **Frequency spectra of FPAnet and VD-Moire demoiréing**

# O  Visual Analysis of Moiré Pattern in Generative Methods

The figures below illustrate the impact of Moiré patterns across generative methods. Figure 27 compares the frequency characteristics of GAN and Non-GAN images, while Figure 28 shows Moiré patterns in images from six generative methods.

## O.1  Moiré Pattern on GAN vs Non-GAN Generated Datasets

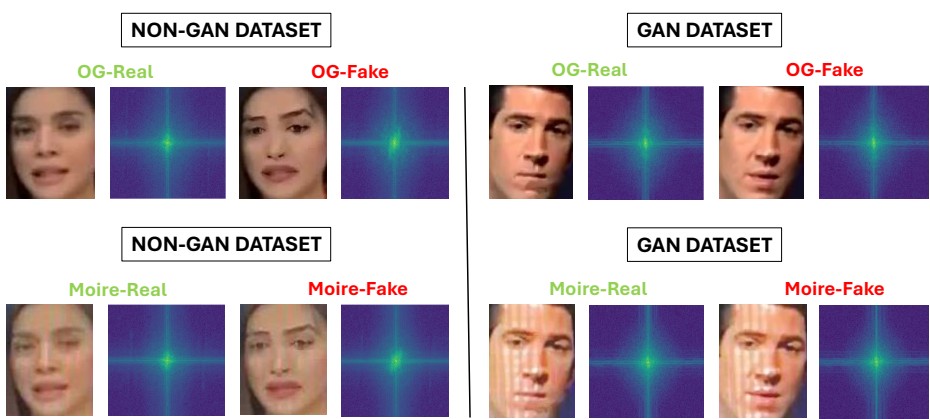

Figure 27: **GAN and Non-GAN images exhibit distinct characteristics under frequency analysis, with Moiré patterns displaying significantly different low-frequency patterns compared to the original images in both datasets.**

## O.2  Moiré Pattern on Multiple Generative Methods

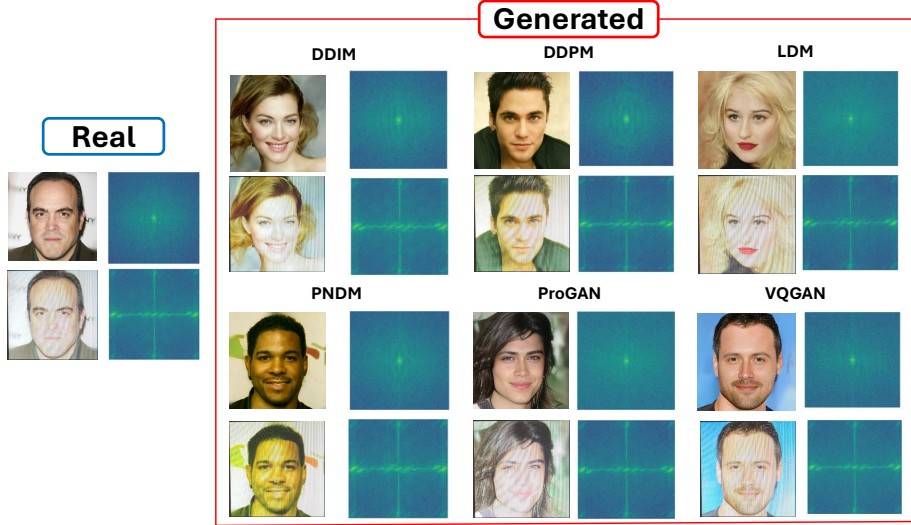

Figure 28: **Images from six generative methods were captured using the Samsung S22 Plus camera, revealing the presence of Moiré patterns. Frequency analysis of these Moiré-impacted images indicated significant differences compared to their original counterparts.**

# P Influence of Side-Angle Captures on Visual Artifacts and Model Performance

We simulate realistic scenarios in which malicious actors record deepfake content from screens using smartphones. Videos were captured from three different angles to introduce authentic Moiré patterns, as shown in Figure 29. Specifically, recordings were taken from a fixed position at a 45° angle from the left (a), a 45° angle from the right (b), and a dynamically moving handheld view (c). This process yielded a total of 12 new videos (6 real, 6 fake) for each of the CelebDF, DFD, DFDC, and UADFV datasets, and 24 new videos (12 real, 12 fake) for the FF++ dataset, reflecting plausible user behaviors and generating diverse real-world distortions.

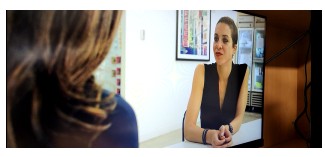 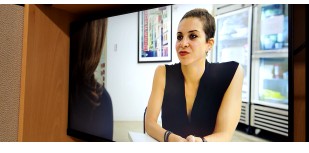 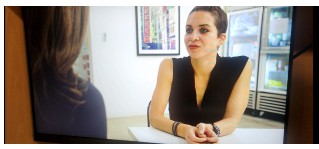

(a) Left 45° view       (b) Right 45° view       (c) Handheld view

Figure 29: **Example frames captured from two fixed angles and handheld view. Each showing cropped views of moiré-affected screen content.**

## P.1 Performance on Moiré Data Captured from ±45° Viewing Directions

Table 15 summarizes the performance of both image-based and video-based detectors under authentic moiré conditions induced by varying mobile capture angles. Overall, most detectors exhibited performance degradation under Moiré distortions. Among video-based models, LipForensics demonstrated the highest resilience, maintaining AUC above 96% under both 45° angle scenarios, although its performance declined under the more dynamic handheld setting. These observations highlight the complex interplay between real-world artifacts and detector behavior, underscoring the need for comprehensive evaluations beyond ideal clean conditions.

Table 15: **Detection AUC of image- and video-based detectors under various moiré pattern conditions. We evaluate four moiré scenarios: LEFT 45°, RIGHT 45°, and handheld captures, in addition to the original (clean) setting. Bold indicates the best performance in each row.**

|  | DETECTORS | ORIGINAL | ANGLE | | |
|---|---|---|---|---|---|
|  |  |  | MOIRE LEFT 45 | MOIRE RIGHT 45 | HANDHELD |
| **IMAGE** | Rossler c23 | **90.6** | 66.1 | 71.4 | 60.0 |
|  | MAT | 71.9 | 73.2 | **82.1** | **82.9** |
|  | CADDM | 78.1 | **76.8** | 78.6 | 51.4 |
| **VIDEO** | Altfreezing | **100.0** | 96.4 | 75.0 | 59.4 |
|  | FTCN | 56.3 | 62.5 | 68.8 | 65.6 |
|  | LipForensics | **100.0** | **96.9** | **96.9** | **84.4** |

# Q License

Our dataset is available to download for non-commercial research and education purposes under a Creative Commons Attribution 4.0 International License. To ensure responsible use of publicly available data, we carefully reviewed the licensing terms of all source datasets used in constructing DeepMoiréFake and contacted the original authors. We have taken appropriate steps to address any potential copyright concerns and designed our distribution process to comply with the varying terms and conditions of the underlying datasets.