# OpenReview forum: "Through the Lens: Benchmarking Deepfake Detectors Against Moiré-Induced Distortions"
_NeurIPS.cc/2025/Datasets_and_Benchmarks_Track — NeurIPS 2025 Datasets and Benchmarks Track poster_

### Official Review · Reviewer_Ax4L · 2025-07-02

**Rating:** 5
**Confidence:** 4

**Summary:**

This paper introduces a new 12,832-video DMF dataset of real and synthetic Moiré-distorted screen captures and systematically evaluates 15 state-of-the-art deepfake detectors, revealing over 20% accuracy drops and the ineffectiveness of demoiréing methods.

**Dataset Code Accessibility:**

Partly

**Ethical Considerations:**

No, there are no or only very minor ethics concerns

**Final Justification:**

After reading the author's rebuttal, most of my questions on Moiré artifacts, device-specific distortions, and data collection are resolved, so I’ve decided to raise my score.

**Limitations Weaknesses:**

As mentioned in the abstract, “deepfake detectors on Moiré-affected videos—an issue that has received little attention.” This topic draws little attention maybe also because most Moiré artifacts arise from demosaicing using the widely used Bayer or non-Bayer filter patterns in mobile-phone cameras. This means that, if a better demosaicing method can alleviate the issue, we wouldn’t need a Moiré-robust detector. So this is like no one jumps into a lake in winter—there must be a reason.

The dataset-capture setup seems somewhat odd: as stated, “They are captured with four different computer screens using two different smartphone cameras under two different lighting conditions on videos from FaceForensics++ (FF++) [23], Celeb-DF [19], the DeepFake Detection (DFD) [24], the DeepFake Detection Challenge (DFDC) [18], and UADFV [20].” Does relying on smartphone cameras to film computer screens—rather than using direct digital screen captures—truly reflect realistic Moiré scenarios, or does it introduce device-specific distortions that may not generalize across varied real-world setups? In addition, using a camera also introduces extra Moiré artifacts when the ISP’s demosaicing is applied.

How were the rael-world screen-capture videos collected and annotated to ensure the Moiré artifacts and ground-truth labels accurately reflect diverse smartphone and display conditions?

And, given that both synthetic Moiré generation and demoiréing methods degrade performance across 15 SOTA detectors, what statistical analyses support these findings and what concrete strategies do the authors recommend for building Moiré-robust deepfake detectors?

**Strengths Contributions:**

A Moiré Pattern-impacted deepfake dataset is collected

The paper is well-written, reader friendly.

---

> ### Author Rebuttal · Authors · 2025-07-31
>
> We thank the reviewer for their insightful comments, and we would like to clarify these concerns as follows:
>
> **Q1.** most Moiré artifacts arise from demosaicing…
>
> **A1.** Thank you for the suggestion. While better demosaicing could hypothetically reduce Moiré formation at capture time, our work is grounded in the practical reality of operating on Moiré-affected content that already exists, where such hardware-level improvements are neither retroactively applicable nor universally deployed. We will include this important consideration in our revised manuscript to emphasize the real-world constraints that motivate our approach.
>
> **Q2.** Does relying on smartphone cameras to film computer screens…
>
> **A2.** Thank you for your comment. Although digital screen capturing is technically available, real-world user behavior frequently favors the convenience of filming screens using smartphones, especially in casual sharing or unauthorized reproduction. This tendency is further driven by DRM (Digital Rights Management) restrictions on many platforms.
>
> We will include this observation in the revised manuscript to better contextualize the prevalence of smartphone-recorded screen content.
>
> **Q2.1.** device-specific distortions
>
> **A2.1.** Thank you for the thoughtful comment. While we agree that device characteristics such as sensor type and screen technology can influence the appearance of Moiré artifacts, we would like to clarify that Moiré formation is not strictly determined by device-specific properties. Based on our observations, factors such as camera-to-screen distance, viewing angle, screen resolution, and ambient lighting conditions often have a more substantial impact on the visibility and intensity of Moiré patterns, even when the same device is used.
> This understanding informed our decision to systematically vary these contextual factors during data collection. Although the number of devices used in our study was limited, we focused on capturing a broad range of realistic viewing conditions that better reflect how Moiré distortions occur in practical scenarios. We believe this approach provides a more meaningful basis for evaluating the robustness of detection models under diverse capture settings. We appreciate the opportunity to clarify this point and will revise the manuscript to better explain the rationale behind our data collection strategy.
>
> **Q3.** How were the real-world screen-capture videos collected and annotated…
>
> **A3.** We appreciate the reviewer’s attention to data quality. To construct the dataset, we first selected a subset of videos from five widely used public deepfake benchmarks to ensure broad diversity in generation techniques and content. Specifically, from the FaceForensics++ dataset, we randomly sampled 50 real and 50 fake videos from each of its four sub-datasets, resulting in a total of 400 videos. For DFD, DFDC, Celeb-DF, and UADFV, we selected one real and one fake video per unique identity, yielding 56, 132, 116, and 98 videos respectively. In total, we curated 802 videos across the five datasets. This sampling strategy was designed to balance diversity with feasibility, as the creation of display-camera captured videos requires manual handling for each video.
> After selecting the videos, we performed the capturing process using various screen and smartphone combinations to induce realistic Moiré artifacts. During this stage, we ensured label accuracy first through automated process followed by careful manual verification and maintained consistency across different screen and camera setups. Furthermore, we categorized the dataset based on key capturing conditions, including device type, display screen, viewing angle, and lighting environment. These attributes are systematically documented and reflected in the metadata of the DeepMoiréFake dataset, supporting reproducibility and facilitating further analysis.
>
> **Q4.** both synthetic Moiré generation and demoiréing methods degrade performance…
>
> **A4.** We thank the reviewer for their observation. As shown in Table 10 of the Appendix, post-processing techniques such as demoiréing, deblurring, and denoising can indeed degrade the performance of existing deepfake detectors. While the primary scope of our work is to evaluate current detector robustness, we acknowledge this limitation and suggest future directions that incorporate training strategies explicitly designed to handle real-world distortions. In particular, leveraging datasets like DeepMoireFake, which include a diverse range of environmental artifacts and degradations, may help enhance model resilience to such conditions. This point will be added to the revised manuscript to guide future improvements in detector robustness.

---

> > ### Comment · Reviewer_Ax4L · 2025-08-05
> > **Response to rebuttal**
> >
> > Thank you for the effort in providing a detailed rebuttal; after reading it, most of my questions on Moiré artifacts, device-specific distortions, and data collection are resolved, so I’ve decided to raise my score. Good luck!

---

> > > ### Author Response · Authors · 2025-08-08
> > >
> > > We are truly grateful for your time and effort in carefully reviewing our rebuttal and for letting us know that it addressed your questions on Moiré artifacts, device-specific distortions, and data collection.
> > >
> > > We sincerely appreciate your decision to raise your score, as well as your encouraging words. Your constructive feedback during the review process has been invaluable in helping us refine and strengthen our work, and we are committed to reflecting these improvements in the final version.

---

### Official Review · Reviewer_hZdE · 2025-07-02

**Ethics Flags:** Data privacy, copyright, and consent,…
**Rating:** 4
**Confidence:** 3

**Summary:**

This paper investigates a critical and practical problem in deepfake detection: the performance degradation of detectors when faced with Moiré-pattern-induced distortions. This issue commonly arises when deepfake videos are recaptured from digital screens using cameras. The authors introduce a new, large-scale benchmark dataset called DeepMoiréFake (DMF), created by systematically re-recording over 12,000 videos from popular deepfake datasets under diverse real-world conditions. They conduct extensive experiments, evaluating 15 state-of-the-art (SOTA) detectors on the DMF dataset. A key finding is that Moiré artifacts can reduce detection accuracy by up to 25.4%. They also find that applying demoiréing techniques, a potential mitigation, further worsens performance, highlighting a significant challenge for current detection systems.

**Additional Feedback:**

The layout of the article is a bit confusing, with tables and images embedded in the text, so I hope the author can optimize the layout.

**Dataset Code Accessibility:**

Yes

**Dataset Code Comments:**

The authors have provided clear and accessible links for their resources. Footnotes 1 and 2 on page 2 provide a DOI for the dataset and a GitHub link for the code. The paper and its extensive appendix contain detailed descriptions of the experimental setup, data collection process (Tables 1 & 2), and evaluation metrics.

**Ethical Considerations:**

Yes, there are ethics concerns that require attention by the authors

**Final Justification:**

The rebuttal mildly addressed my concerns (no extra experiments or even benchmark designs are involved). I will leave my score unchanged.

**Limitations Weaknesses:**

- **Self-Acknowledged Limitations:** The authors commendably acknowledge in their discussion (Section 6, "Limitation and Future work") that their study does not cover all possible variations, such as the full range of camera/display hardware or dynamic motion artifacts. This is a reasonable scope limitation for a single paper.

- **Deeper Analysis of Mitigation Failure:** While the paper excellently demonstrates that demoiréing fails, the explanation could be more in-depth. It is hypothesized that demoiréing removes critical deepfake artifacts along with Moiré patterns (Section 5, lines 385-388). A deeper feature-level analysis or visualization could provide stronger evidence for why this happens, which would be highly valuable for designing future solutions.

- **Limited Exploration of Adaptive Solutions:** The paper focuses on benchmarking existing detectors. While fine-tuning and retraining are explored, there is less focus on proposing a new, adaptive solution designed specifically to be robust against Moiré patterns.

**Strengths Contributions:**

- **Novelty:** This paper systematically study the "screen-to-camera" deepfake problem, a critical but overlooked real-world challenge. It shifts research from clean datasets to noisy, practical scenarios.

- **Rigorous Evaluation:** The experiments are highly thorough, featuring a new, large-scale dataset (DMF) with diverse conditions and a benchmark of 15 state-of-the-art detectors.

- **Impactful Findings:** The key discovery is that removing Moiré patterns (demoiréing) unexpectedly **worsens** detection performance. This counter-intuitive result challenges current assumptions and opens new research directions.

- **Quality and Contribution:** The paper is well-written and clear. By releasing their dataset and code, the authors provide a valuable resource that will drive future research.

---

> ### Author Rebuttal · Authors · 2025-07-31
>
> We thank the reviewer for their insightful comments, and we would like to clarify these concerns as follows:
>
> **Q1.** Self-Acknowledged Limitations…
>
> **A1.** We thank the reviewer for this valuable suggestion. Although our experiments were designed to reflect everyday usage scenarios, the scope of the current work focused on evaluating deepfake detection across the most commonly used devices that are frequently encountered in diverse real-world settings. We acknowledge the importance of expanding this scope to include a broader range of device types and will consider incorporating such diversity in future work.
>
> **Q2.** Deeper Analysis of Mitigation Failure..
>
> **A2.** We thank the reviewer and would like to clarify the reasoning. In our experiments, we conducted an spectral analysis to understand the reasons behind the limited success of the demoiréing process, as illustrated in Figure 27 in the Appendix. Our findings indicate that when a video is captured particularly through a camera the original moiré patterns overwrite or obscure some of the underlying deepfake artifacts present in the video. As a result, this information is partially lost during the capture process. Consequently, even when demoireing is applied, it cannot fully recover the original content, since the deepfake-relevant signals have already been degraded by the moiré interference.
>
> **Q3.** Limited Exploration of Adaptive Solutions…
>
> **A3.** Thank you for the thoughtful comment. In this study, we aimed to assess the robustness of existing deepfake detectors under a relatively underexplored but practically important distortion, Moiré artifacts that occur during display-camera capture. To facilitate this, we introduced the DeepMoiréFake dataset, which allows for systematic benchmarking under these real-world conditions. Our goal is to provide a useful benchmark and diagnostic analysis to better understand how current models behave under these real-world perturbations. We used existing popular defense adaptive approaches such as Image demoiréing (ESDNet, MBCNN, DMCNN, DDA) and Video demoiréing (Video Demoiréing, and FPANet) to remove the moire pattern. The details of these experiments are provided in appendix Sections E and F, respectively.
> While we do not propose a new adaptive solution in this study, we hope that the findings can serve as a stepping stone for future efforts aimed at developing detection models that are more resilient to Moiré distortions. We agree that exploring approaches such as Moiré-invariant architectures or domain-adaptive techniques would be a valuable direction for future research, and we hope that the insights and resources provided in this paper can help support such advancements.
>
> **Q4.** Data privacy, copyright, and consent, Safety and security…
>
> **A4.** Thank you for raising this important point. We take ethical considerations seriously and have implemented appropriate safeguards in the release of our dataset. As detailed in Section Q of the appendix, our dataset is distributed under a Creative Commons Attribution 4.0 International License and is available strictly for non-commercial research and educational purposes.
> To ensure responsible use, we carefully reviewed the licensing terms of all source datasets used in constructing DeepMoiréFake and contacted the original authors where necessary. We have taken appropriate steps to address any potential copyright concerns and designed our distribution process to comply with the terms and conditions of the underlying datasets. These measures aim to ensure ethical and lawful use of the dataset within the research community.

---

> > ### Comment · Reviewer_hZdE · 2025-08-08
> >
> > The authors didn't offer any further complementary experiments or additional designs in their rebuttal to fully address my concerns regarding the devices' diversity in real-world application scenarios. I will keep my score.

---

> > ### Author Response · Authors · 2025-08-08
> >
> > We sincerely thank the reviewer for reiterating this important point and for highlighting the value of incorporating a broader diversity of devices in our evaluation.
> >
> > We carefully considered the reviewer's thoughtful suggestion during the rebuttal stage. However, according to the conference policy, we were not allowed to include totally new experimental results. So, I hope the reviewer understands the fact, and our  circumstances, and kindly take those into consideration.
> >
> > That being said, however, we fully agree that expanding the scope to cover a broader range of camera/display hardware would further strengthen the real-world applicability of our findings. For the final version, we plan to include complementary experiments requested by the reviewer and share it on our GitHub repository to benefit the research community.

---

> > > ### Comment · Reviewer_hZdE · 2025-08-09
> > >
> > > Thanks for your commitment on the furture development of your work, and wish you good luck.

---

### Official Review · Reviewer_o7nG · 2025-07-02

**Rating:** 4
**Confidence:** 3

**Summary:**

This research paper presents a significant investigation into the impact of Moiré patterns—visual distortions arising when recording digital screens with smartphones—on the performance of state-of-the-art deepfake detectors. The authors introduce the DeepMoiréFake (DMF) dataset, comprising 12,832 videos (35.64 hours) captured under real-world conditions (varying screens, smartphones, lighting, angles) from established datasets (Celeb-DF, DFD, DFDC, FF++, UADFV). They rigorously evaluate 15 top detectors across three attack scenarios: Captured Moiré Pattern Attack (CMPA) (authentic distortions), Synthetic Moiré Pattern Attack (SMPA), and Compression Attack (CA). Key findings include severe performance degradation (up to 25.4% drop in AUC for CMPA, 21.4% for SMPA), exacerbated by compression. Surprisingly, demoireing techniques intended as mitigation worsened accuracy by up to 16%, inadvertently removing crucial deepfake artifacts. The work underscores the vulnerability of current detectors to real-world distortions and releases DMF (via controlled DOI access) to spur robust detection research.

**Dataset Code Accessibility:**

Yes

**Ethical Considerations:**

No, there are no or only very minor ethics concerns

**Limitations Weaknesses:**

1. Experiments used only LED/LCD (LG, BenQ) and IPS (Samsung, Lenovo) monitors. Newer display technologies like OLED (noted as less prone to Moiré) or high-refresh-rate/8K screens were excluded, despite being mentioned as future work. This restricts understanding of Moiré impact across modern devices.
2. Capture relied solely on iPhone 13 and Samsung S22+. Given the observed performance difference between these devices (Samsung generally worse), broader testing across budget/mid-range phones, varied camera sensors (e.g., Sony vs. Samsung ISOCELL), and OS-level image processing pipelines is needed to assess real-world generalizability.
3. Results showed minimal performance change (<1%) between "light on" and "light off" conditions. This suggests the tested lighting variations were insufficiently challenging or that Moiré distortion dominates lighting effects, but deeper investigation into extreme/low-light scenarios common in real recordings is lacking.
4. While demoireing was extensively tested (showing negative results), other promising mitigations like domain adaptation, robust feature learning, or Moiré-invariant detector architectures were not deeply explored. Fine-tuning/retraining improved compressed data performance but wasn’t systematically applied to the core Moiré distortion problem.
5. Videos were captured at a fixed distance (35cm) and primarily static angles (handheld variation mentioned but not deeply analyzed). Real-world scenarios involve motion, varying distances, and unstable camera handling, potentially exacerbating Moiré artifacts—factors not modeled.
6. Compression attacks used only H.264 (C23/C40). Real social media platforms employ diverse, often proprietary, compression algorithms (e.g., TikTok/Instagram’s aggressive bitrate reduction + filters), and their interaction with Moiré could differ significantly.

**Strengths Contributions:**

The study addresses a critical, under-explored gap in deepfake detection robustness. The creation of the DMF dataset is a major contribution, providing the first benchmark incorporating authentic Moiré distortions under controlled, diverse real-world conditions. The experimental design is thorough, evaluating a wide array of detectors (15 models) across multiple distortion scenarios (CMPA, SMPA, CA) and mitigation strategies (demoireing, fine-tuning, retraining). The discovery that demoireing harms detection performance is counterintuitive and valuable, highlighting unintended consequences of preprocessing. The paper acknowledges societal impact, implements dataset safeguards (DOI access control), and provides detailed reproducibility.

---

> ### Author Rebuttal · Authors · 2025-07-31
>
> We thank the reviewer for their insightful comments, and we would like to clarify these concerns as follows:
>
>
> **Q1.** Experiments used only LED/LCD (LG, BenQ) and IPS (Samsung, Lenovo) monitors…
>
> **A1.** Thank you for this thoughtful comment. The experimental setup was intentionally designed around commonly used consumer devices, including popular display types and widely adopted smartphones.
> We agree, however, that evaluating newer display technologies such as OLED or 8K monitors, as well as a broader range of mobile devices with different camera sensors and processing pipelines, would offer a more comprehensive understanding of Moiré artifacts and their impact. As noted in the manuscript, we consider this an important direction for future work and plan to expand our evaluation accordingly.
>
> **Q2.** Moiré distortion dominates lighting effects…
>
> **A2.** Thank you for your comment. We would like to clarify that the "lights off" condition in our experiments was intended to represent an extreme low-light scenario. Specifically, these captures were conducted at night, after complete sunset, with no ambient room lighting. In this setting, the only illumination came from the display screen itself, which was recorded by the smartphone camera. The minimal performance difference observed between "light on" and "light off" conditions suggests that Moiré distortions are dominant enough to overwhelm the variations introduced by lighting changes. In other words, Moiré patterns appear to be almost equally effective in degrading detection performance whether the lights are on or off.
>
> **Q3.** Other promising mitigations…
>
> **A3.** We thank the reviewer for these insightful suggestions. Our study focused on direct mitigation strategies such as fine-tuning, retraining, and video-based demoireing. We agree that other approaches, including domain adaptation, robust feature learning, and Moiré-invariant architectures, are promising directions that could further improve model robustness. While these were not explored in the current study, we consider them valuable avenues for future research.
> With the release of our benchmark dataset, we hope it can provide a useful foundation for the community to explore and develop future detection models that address these challenges more effectively. We look forward to seeing future work build upon this dataset to advance robustness against Moiré artifacts.
>
> **Q4.** but wasn’t systematically applied to the core Moiré distortion problem…
>
> **A4.** Thank you for your comment. We would like to clarify in case there was any misunderstanding caused by how we explained this aspect in the manuscript. In our study, we conducted retraining and fine-tuning experiments to examine model robustness against Moiré distortions. As shown in Table 8, we included results for models retrained or fine-tuned using various types of Moiré-affected data, including authentic patterns (CMPA), two synthetic variants (SMPA1 and SMPA2), and the original data. These experiments were intended to explore how different types of Moiré distortions influence detection performance under retraining or fine-tuning. We appreciate the opportunity to clarify this and will revise the text to communicate the scope and intent of our evaluation more clearly.
>
> **Q5.** Videos were captured at a fixed distance (35cm) and primarily static angles…
>
> **A5.** Thank you for your insightful comment. We acknowledge the relevance of slight variations in handheld usage, which is why we incorporated handheld variation during data collection to partially reflect real-world instability. Additional experiments analyzing the effect of handheld captures and different viewing angles are included in Appendix Section P to provide further insight into these variations. We also recognize the importance of exploring a wider range of viewing distances. However, due to the significant human labor that would have been required to systematically combine multiple viewing distances with other variations, we were unable to fully explore this aspect. As a practical alternative, we adopted handheld capture settings to introduce natural variation in both distance and angle, aiming to better approximate real-world usage scenarios. We appreciate the suggestion and agree that exploring more dynamic capture scenarios could be an interesting direction for future work.
>
> **Q6.** Compression attacks used only H.264 (C23/C40)...
>
> **A6.** We thank the reviewer for highlighting this point. Various social media platforms employ different video compression algorithms, with H.264 being among the most commonly used on platforms such as TikTok, Instagram, and YouTube. To ensure a realistic representation of compression artifacts found in social media videos, we incorporated H.264 compression in our testing pipeline. This choice was made to align our evaluation with typical platform-specific video processing and enhance the practical relevance of our findings. However there are other proprietary algorithms are not open-sourced and couldn’t incorporate them in our current study but will be adapting them in future research.

---

### Official Review · Reviewer_bxnm · 2025-07-03

**Rating:** 4
**Confidence:** 4

**Summary:**

- This paper highlights the issue of Moiré patterns that arise when deepfake content is re-captured from screens, emphasizing their impact as an adversarial attack on deepfake detectors. To faciliate further research, authors introduce a new benchmark dataset designed to evaluate detectors under such conditions.
- The proposed dataset consists of two components: a real-world setting, where existing deepfake videos are recorded using smartphones under various environments, and a synthetic setting where Moiré patterns are artifically generated using two different methods.
- Using the proposed dataset, the authors perform a comprehensive evaluation ad bencmarking of existing deepfake detection models in the presence of Moiré artifacts

**Dataset Code Accessibility:**

No

**Dataset Code Comments:**

Authors mentioned that dataset will be released upon acceptance

**Ethical Considerations:**

No, there are no or only very minor ethics concerns

**Final Justification:**

Some of the issues I raised—specifically the use of synthetic data (Q1) and the exclusion of certain baseline methods (Q2)—were sufficiently addressed in the authors’ response. I believe these points should be clearly incorporated into the paper, as omitting them would break the flow for readers.

Although Q3 and Q4 were also explained in the rebuttal, the fact that these clarifications were not included in the original submission raises concerns about the overall reliability of the experimental section.

Given the space constraints of the NeurIPS format, I believe the authors need to more carefully consider which content belongs in the main text versus the appendix, and which points require explanation versus those that can be summarized.

Taking other reviewers’ feedback into account, my final recommendation is borderline accept. However, due to concerns about the completeness and polish of the manuscript, my evaluation is slightly more reserved compared to other reviewers.

**Limitations Weaknesses:**

Q1. Fine-tuning / retraining on synthetic datasets
- The experimental setup for fine-tuning and retraining is not described in sufficient detail. Please specify the exact settings of data were used for training and were used for evaluation.
- Additionally, if the same setup of synthetic Moiré patterns are applied during both training and testing, then the results may not be meaningful, as the model would be learning pattern-specific features rather than general robustness. To properly evaluate effectiveness, the model should be trained on synthetic data and tested on real Moiré examples only.

Q2. Clarification on Line 275 - Performance of SelfBlended and CCViT
- Could the authors explain why SelfBlended and CCViT failed to show performance improvement, or even suffered performance degradation, during fine-tuning?
- If such high variance or collapse frequently occurs during fine-tuning, it would suggest that the synthetic dataset may not be consistently beneficial for learning.
- This raises concerns about the usefulness and reliability of the proposed dataset.

Q3. Clarification on Line 348 - SMPA Results
- In the SMPA experiment, some settings seem to outperform the original "no-attack" baseline, while performance trends across models appear inconsistent.
Could the authors explain why MAT shows a significant performance drop, whereas Rossler C23 and CADDM even show improvements in SMPAs?
- Based on these results, it is unclear whether the synthetic experiments under SMPA are practically meaningful.
If there is low consistency between SMPA and CMPA results, it may indicate issues in the SMPA generation methodology.
Alternatively, the performance degradation might stem for factors beyond Moiré patterns, which should be investigated.
- Please provide a complete table of results for all 15 models.

Q4. Clarification on Line 360 - Compression Attack Results
- The values "accuracy of 74.4%, 52.2%, and 72.3%" mentioned in Line 360 do not appear in Table 7 or any other table. Could the authors clarify where these numbers were derived from?
- Also, the section gives the impression that the discussion goes beyond the intended scope of the proposed benchmark dataset targetting Moiré patterns.

**Strengths Contributions:**

- The paper extends prior work by systematically analyzing and quantifying the effects of Moiré patterns, providing a deeper understanding of a real-world distortion that has often been overlooked in previous deepfake detection research.
- The proposed dataset is well-structured, incorporating multiple screen types, lighting conditions, and camera angles, which enables robust benchmarking across realistic scenarios
- The authors conduct extensive experiments using 15 state-of-the-art detection models.

---

> ### Author Rebuttal · Authors · 2025-07-31
>
> We thank the reviewer for their insightful comments, and we would like to clarify these concerns as following.
>
> **Q1.** Fine-tuning / retraining on synthetic datasets…
>
> **A1.** We thank the reviewer for their comment. In our approach, we used the original dataset and captured the moire dataset as primary Fine-tune and retrain data, and to evaluate the trained model, we selected the Original, Moire, and synthetically generated datasets. (SMPA and SMPA-MA). This ensured that the trained model would be evaluated to unseen artifacts not seen during the training enabling a more robust assessment of generalization to real-world distortions. We will include the specific settings of  dataset composition and experimental settings to improve clarity and reproducibility.
>
> **Q2.** Clarification on Line 275 - Performance of SelfBlended and CCViT…
>
> **A2.** Thank you for your thoughtful comment regarding the performance of SelfBlended and CCViT under retraining and fine-tuning.
> For CCViT, the performance degradation during fine-tuning can be attributed to two main factors. First, CCViT employs a relatively large transformer-based architecture, which typically requires more training epochs and careful learning rate scheduling to converge effectively. In our experiments, we maintained a fixed number of training epochs across all models for fairness and comparability. However, lighter-weight models such as Xception or F3Net were able to converge within this budget, while CCViT's training likely remained under-optimized due to its slower convergence rate. Second, transformer-based models like CCViT are known to be more sensitive to hyperparameter settings and data distribution shifts. [1] Fine-tuning on limited or domain-shifted data (such as Moiré-affected samples) without model-specific adaptation strategies may exacerbate overfitting or misalignment in attention, leading to a performance drop.
> In contrast, we would like to clarify that the SelfBlended method is fundamentally different from standard detection models, as it relies on specifically crafted self-blended data [2] during training. Therefore, applying conventional retraining or fine-tuning procedures is not directly applicable for this approach. Evaluating it under standard retraining settings would not align with its intended use or training paradigm. We will revise the manuscript accordingly to clearly reflect these distinctions and prevent any potential confusion in the final version.
>
> **Q3.** Clarification on Line 348 - SMPA Results…
>
> **A3.** Thank you for your thoughtful question. The performance drop observed in MAT under SMPA (Synthetic Moiré Pattern Attack) conditions can be attributed to its architectural characteristics. Unlike Rossler and CADDM, MAT relies on an attention-based mechanism to guide feature extraction. While attention modules can help models focus on relevant regions, they are also known to be sensitive to input perturbations.
> Recent studies have shown that attention mechanisms are particularly vulnerable to both adversarial and incidental disturbances. For instance, Liong et al. [3] demonstrate that even slight perturbations in attention scores, without modifying the input itself, can lead to a substantial drop in model performance. Their findings indicate that a 1% change in attention weights can result in a 98% attack success rate, emphasizing how easily attention mechanisms can be disrupted.
> In our case, MAT may be especially sensitive to synthetic moiré patterns, which were not part of its training distribution. These artifacts likely interfere with the learned attention maps, leading the model to focus on irrelevant or misleading regions. Such disruption in attention could explain the considerable performance degradation observed under SMPA conditions. We appreciate the reviewer’s observation and will revise the manuscript to include this explanation for greater clarity.
>
> **Q3.1.** it is unclear whether the synthetic experiments under SMPA are practically meaningful…
>
> **A3.1.** Thank you for your valuable comment. While the results reveal a discrepancy between SMPA (Synthetic Moiré Pattern Attack) and CMPA (Captured Moiré Pattern Attack), we believe this gap is highly meaningful. It demonstrates that synthetic moiré patterns fail to fully capture the complex and diverse characteristics of real, camera-captured moiré artifacts. This limitation underscores the importance of using authentic moiré data when evaluating model robustness.
> This observation directly motivates the introduction of our DMF dataset, which contains authentic moiré distortions collected through real-world display-camera capture pipelines. Without such real data, evaluations based solely on synthetic perturbations may offer an incomplete or even misleading view of detector performance in practical scenarios. Thus, the inconsistency between SMPA and CMPA highlights the value and necessity of incorporating authentic moiré patterns for realistic and comprehensive robustness assessments.
>
> **Q4.** Clarification on Line 360 - Compression Attack Results…
>
> **A4.** Thank you for pointing this out. We would like to clarify that all experiments involving retraining and fine-tuning were thoroughly conducted across all relevant settings. Specifically, we evaluated RosslerC23, MAT, and CADDM on four data types (Original, CMPA, SMPA-MA, and SMPA-SPS) under both C23 and C40 compression levels. For each combination, we collected and verified baseline as well as fine-tuned performance results.
> In the manuscript, we referred to these baseline values when describing the effect of retraining and fine-tuning. However, the specific numbers were not explicitly listed in the corresponding table, which may have caused some confusion. We confirm that all baseline results associated with the settings in Table 7 were computed and used consistently throughout the analysis. To enhance clarity and completeness, we will revise the manuscript to ensure that these values are clearly presented or properly referenced in the text. We appreciate the opportunity to clarify this and hope it reinforces the reliability and transparency of our experimental evaluation.
>
> **Q4.1.** the section gives the impression that the discussion goes beyond the intended scope…
>
> **A4.1.** The reason for doing compression attack was to emulate the Real-World scenario where posts uploaded to social media goes degradetion due to compression algorithms deployed by social media platforms, and the experiments results show the effect of Moire pattern in combination with compression on Deepfake Detectors.
>
> We appreciate the reviewer’s observation. The incorporation of compression attacks in our study was intentional and aimed at emulating real-world scenarios in which user-uploaded content undergoes degradation due to platform-specific compression algorithms. Our goal was to assess how Moiré patterns interact with such compression artifacts, as commonly encountered on social media platforms. The experimental results highlight that this combination can significantly impact the performance of deepfake detectors. While the primary scope remains focused on evaluating detector robustness, we believe this added dimension enhances the practical relevance of our findings.
>
> **References:**
>
> **[1].** Wiemerslage, Adam, Kyle Gorman, and Katharina von der Wense. "Quantifying the hyperparameter sensitivity of neural networks for character-level sequence-to-sequence tasks." Proceedings of the 18th Conference of the European Chapter of the Association for Computational Linguistics (Volume 1: Long Papers). 2024.
>
> **[2].** Shiohara, Kaede, and Toshihiko Yamasaki. "Detecting deepfakes with self-blended images." Proceedings of the IEEE/CVF conference on computer vision and pattern recognition. 2022.
>
> **[3].** Liong, Khai Jiet, Hongqiu Wu, and Hai Zhao. "Unveiling Vulnerability of Self-Attention." arXiv preprint arXiv:2402.16470 (2024).

---

> > ### Comment · Reviewer_bxnm · 2025-08-07
> >
> > Thank you for the authors’ detailed and thoughtful response. After going through the rebuttal and revisiting the paper, I found the overall structure to be well thought out and carefully designed.
> >
> > That said, I still believe the final camera-ready version should address some of the points I raised—particularly those related to experimental design, model selection, and interpretation of the results.
> >
> > I’d also like to mention readability. If space is tight under the NeurIPS format, I encourage you to make full use of the appendix, rather than omitting important connections or explanations. Additionally, clearer placement of figures and tables would help improve the overall flow and clarity of the paper.
> >
> > Wishing you all the best with the final version.

---

> > > ### Author Response · Authors · 2025-08-08
> > >
> > > We sincerely thank you for taking the time to revisit our paper after the rebuttal and for your encouraging feedback on the overall structure and design.
> > >
> > > We truly appreciate your acknowledgment of our contributions and your suggestions regarding experimental design, model selection, result interpretation, and overall readability are invaluable. We will carefully incorporate these points in the camera-ready version, making full use of the appendix where necessary, and we will work to improve the clarity and placement of figures and tables to ensure a smoother reading experience.

---

### Note · Authors · 2025-08-14

We thank AC, SAC, and reviewers for their constructive and thoughtful feedback. We appreciate that the core idea was recognized as **relevant** and **timely** (hZdE), the **methodology sound** (bxmn), and the **writing clear** (hZdE, o7nG).

**1. Fairness and Ethics:** Following the ethics reviewer's suggestion, we will include the disaggregated results by gender and race in the appendix of the final version to ensure transparency and equitable analysis.

**2. Real-World Device Diversity.** Reviewer (**hZdE**) recommended expanding device diversity in real-world tests. While new experiments were restricted during rebuttal, we fully agree and will incorporate these results in the final version and release extended benchmarks via GitHub to enhance applicability.

**3. Addressing Unacknowledged Concerns.** Although Reviewer (**o7nG**) did not acknowledge our rebuttal, we believe our clarifications addressed their concerns.

**4. Building on Prior Feedback.** This paper in NeurIPS 2024 Datasets and Benchmarks Track received positive reviews (**6, 7, 8, 7**) with suggestions for broader generalization and distortion coverage [1]. This year’s submission addresses these by:
* Expanding distortions (synthetic Moiré, via screen simulation/attack, compression artifacts, Gaussian blur, and sharpening).
* Integrating advanced video de-Moiré methods (FPANet, VD-Moiré).
* Providing deeper analysis (spectrum difference) of distortion effects and moiré effects on NON-GAN/GAN and Generative methods.
* Additional viewing angles result.

**5. Strengthened Contribution.** These Improvements transform last year’s single-vulnerability observation into a comprehensive, distortion-aware deepfake detection benchmark with practical deployment considerations.

We believe the final version reflects a stronger, clearer, and more impactful contribution aligned with NeurIPS standards.

We appreciate your valuable time and consideration of our submission.

**References:**

**[1]** https://openreview.net/forum?id=d9F394AKDW#discussion

---

### Decision · Program_Chairs · 2025-09-18

**Decision:**

Accept (poster)

**Comment:**

The submission present a dataset that was created by recapturing sequences from standard deepfake detection datasets by two smartphones displayed on  a diverse set of monitors. The process introduces artifacts, most notably Moire patterns. The paper shows in an extensive evaluation that state-of-the-art deepfake detectors are very sensitive to this process and their recognition accuracy drops significantly. The contributions are thus the dataset, the thorough evaluation and some auxiliary experiment.

The reviewers are all in favor of accepting the paper. Their original critical points the reviewer raise are of technical nature, and sometimes of the "we want more of this" (more recording devices, more displays), which are more a sign of interest rather than indicators of a problem. After the rebuttal, most issues are resolved.

The deepfake detection problem raise ethical issues. The authors were aware of this. The ethics review point out some other issues which the authors acknowledge and will address.